# Production of particulate brown carbon during atmospheric aging of residential wood-burning emissions

Nivedita K. Kumar[1], Joel C. Corbin[1*], Emily A. Bruns[1], Dario Massabó[2], Jay G. Slowik[1], Luka Drinovec[3,4], Griša Močnik[3,4], Paolo Prati[2], Athanasia Vlachou[1], Urs Baltensperger[1], Martin Gysel[1], Imad El-Haddad[1] and André S. H. Prévôt[1]

[1]Laboratory of Atmospheric Chemistry, Paul Scherrer Institute, 5232 Villigen, Switzerland

[2]Department of Physics & INFN, University of Genoa, via Dodecaneso 33, 16146, Genova, Italy

[3]Aerosol d.o.o, Kamniška 41, 1000 Ljubljana, Slovenia

[4]Condensed Matter Physics, Jožef Stefan Institute, 1000 Ljubljana, Slovenia

*Now at National Research Council Canada, Ottawa, Canada

*Correspondence to*: I. El-Haddad (imad.el-haddad@psi.ch), A. S. H. Prévôt (andre.prevot@psi.ch)

## ABSTRACT

We investigate the optical properties of light-absorbing organic carbon (brown carbon) from domestic wood combustion as a function of simulated atmospheric aging. At shorter wavelengths (370 – 470nm), light absorption by brown carbon from primary organic aerosol (POA) and secondary organic aerosol (SOA) formed during aging was around 10 % and 20 %, respectively, of the total aerosol absorption (brown carbon plus black carbon). The mass absorption cross-section (MAC) determined for black carbon (BC, 13.7 $m^2 g^{-1}$ (geometric standard deviation GSD = 1.1) at 370 nm) was consistent with that recommended by Bond et al. (2006). The corresponding MAC of POA (5.5 $m^2 g^{-1}$ (GSD =1.2)) was higher than that of SOA (2.4 $m^2 g^{-1}$ (GSD = 1.3)) at 370 nm. However, SOA presents a substantial mass fraction, with a measured average SOA/POA mass ratio after aging of ~5 and therefore contributes significantly to the overall light absorption, highlighting the importance of wood-combustion SOA as a source of

atmospheric brown carbon. The wavelength dependence of POA and SOA light absorption between 370 nm and 660
nm is well described with absorption Ångström exponents of 4.6 and 5.6, respectively. UV-visible absorbance
measurements of water and methanol-extracted OA were also performed showing that the majority of the light-
absorbing OA is water insoluble even after aging.

## 1. INTRODUCTION

Atmospheric aerosols contribute to radiative forcing either directly by absorbing and scattering light or indirectly by
acting as cloud-condensation and ice nuclei. While black carbon (BC) from combustion processes is the most
efficient light-absorbing aerosol component, organic aerosols (OA) may also absorb solar radiation (Alexander et
al., 2008; Chen and Bond, 2009; Kirchstetter et al., 2004). This light-absorbing OA, denoted as brown carbon
(BrC), absorbs most strongly at shorter UV-visible wavelengths (Andreae and Gelencsér, 2006; Hoffer et al., 2005).
Global chemical-transport model estimates indicate that the BrC contribution to the positive radiative forcing of
climate by anthropogenic aerosols may not be negligible (Feng et al., 2013; Jo et al., 2016; Lin et al., 2014; Wang et
al., 2014).
Unlike BC, whose light absorption properties are relatively constant across sources (Bond et al., 2013), BrC is
composed of a wide range of largely unknown compounds, which exhibit highly variable spectral dependence and
absorption efficiencies. For example, reported imaginary indices of refraction for different organic species, which
describe the absorption of these compounds, span two orders of magnitude (Lu et al., 2015). Because it is
impractical to experimentally separate BrC from non-absorbing OA, optical properties are typically determined for
the bulk OA of a given source. The large variability of BrC fraction in combustion aerosol may contribute to the
wide variation in reported properties of BrC containing OA.
Biomass burning OA, which contributes two-thirds of the global budget of directly-emitted primary OA (POA), is
expected to be a considerable source of BrC (Chakrabarty et al., 2010; Hecobian et al., 2010; Lack and Langridge,
2013; Liu et al., 2014). The variability in reported light absorption properties of biomass burning OA with fuel type
and burn conditions remains a major obstacle complicating its treatment in climate models (Lu et al., 2015; Saleh et
al., 2013). Residential biomass burning is typically characterized by a more efficient combustion, than open burning.
Residential wood burning represents a substantial contribution to anthropogenic combustion emissions (Bond et al.,
2013), especially in urban atmospheres, and is considered the largest source of OA in Europe during winter (Denier
Van Der Gon et al., 2015).
Upon photo-oxidation, biomass-burning emissions produce secondary organic aerosol (SOA) at concentrations
similar to or exceeding the primary organic aerosol (POA) (Bertrand et al., 2017; Bruns et al., 2015, 2016; Corbin et
al., 2015a; Grieshop et al., 2009). There is a growing body of evidence that light absorption by OA change with OH
exposure (aging) owing to the production of secondary BrC or to the transformation of primary BrC (Forrister et al.,
2015; Heringa et al., 2011; Lee et al., 2014; Zhao et al., 2015). However, these effects have not yet been
systematically investigated and must be quantified to assess the climate effects of primary and aged biomass burning
OA.
Here, we show that both POA and SOA from residential biomass burning emissions aged in controlled smog
chamber experiments contain BrC. Wavelength dependent, mass-normalized absorption cross-sections (MACs) of
POA and SOA are presented from online aerosol measurements as a function of aging for the first time.
Complementary measurements of filter-extract absorbance (conducted in different solvents) are used to obtain the
imaginary refractive index and to investigate the solubility of BrC in fresh and aged OA. While results presented
here are related to flaming residential wood combustion emissions and cannot therefore be generalized, the approach
used can be extrapolated for the characterization and quantification of the contribution of BrC in other primary and
aged emissions.

**2.  METHODS**
**2.1 Smog chamber experiments**
Laboratory measurements were conducted in an 8 m$^3$ Teflon smog chamber (Bruns et al., 2015; Platt et al., 2013)
installed within a temperature-controlled housing. Conditions in the chamber were maintained to represent winter
time in Europe, i.e. relative humidity ranging between 50 – 90%, at 263 K (Bruns et al., 2015, 2016). Beech wood
was combusted in a residential wood stove. Primary emissions were sampled through heated lines at 413 K, diluted
by a factor of ~14 using an ejector diluter (DI-1000, Dekati Ltd.), then sampled into the chamber, which provided an
additional ten-fold dilution. The overall dilution was a factor of 100 to 200. As we aimed to sample only flaming-
phase emissions into the chamber, samples were taken when the modified combustion efficiency (ratio of $CO_2$ to the
sum of CO and $CO_2$) was > 0.90. Despite maintaining the same combustion conditions, the resulting organic fraction
to the total carbonaceous aerosols in the different samples was highly variable, indicating that these samples are
representative of a mixture of pre- ignition and flaming emissions (with varying contributions of each combustion
stage). Finally, the resulting NOx/NMOG ratios, which dramatically influence SOA formation through influencing
the fate of peroxy radicals, $RO_2$, were estimated to be between $0.035 – 0.35$ ppm ppm $C^{-1}$ (Bruns et al., 2016). These
conditions can be considered as high $NO_X$ representative of urban/sub-urban conditions, where most of the $RO_2$
radicals react with NO, rather with $RO_2/HO_2$.
After injection of the primary emissions and stabilization of the concentrations, nitrous acid (HONO) was
continuously added, which dissociates upon irradiation ($\lambda<400$ nm) and forms the hydroxyl radical (OH). Then, 9-
times deuterated butanol sample (butanol- D9, 98%, Cambridge Isotope Laboratories) was subsequently injected
into the chamber. The decay of butanol-D9 was used to infer the time-resolved OH exposure of the sampled aerosol
(Barmet et al., 2012). The chamber was exposed to UV lights for ~3.5 hours.
Particles were collected onto filters (47 mm Tissue-quartz, Pall Corporation, 26 L $min^{-1}$ for 30-32 min) for offline
optical measurements and the determination of elemental carbon (EC) mass. Three filters were collected during each
experiment, namely i) a primary aerosol filter sample ("primary"), ii) a slightly aged aerosol ("Aged1", OH
exposure ~ $1\times10^7$ molecules $cm^{-3}$ h), collected 30 minutes after the UV lights were switched on, and iii) an aged
aerosol ("Aged2", OH exposure ~ $4\times10^7$ molecules $cm^{-3}$ h), collected at the end of the experiment (see Fig. S1 for
the sampling periods). A charcoal denuder was installed upstream of the filter sampler to remove organic gases.
Filters were stored at 253K until analysis.
In addition to the characterization of the particle optical properties detailed in the next section, a set of online and
offline techniques were used for the characterization of the gaseous and particulate emissions before and after aging.
The non-refractory particle size-segregated chemical composition was measured with a high resolution (HR) time-
of-flight aerosol mass spectrometer (AMS) (DeCarlo et al., 2006). Uncertainties related to particle collection
efficiency in the AMS are considered negligible for the relatively-large particles sampled here, which in terms of
volume are within the size range transmitted efficiently by the AMS aerodynamic lens (Liu et al., 2007). The
collection efficiency of wood-combustion OA is expected to be unity (Corbin et al., 2015b). Details related to the
AMS data analysis and calibration can be found elsewhere (Bruns et al., 2015, 2016). A scanning mobility particle
sizer was used to measure the size distribution of the evolving aerosol. Organic gases were monitored by a proton
transfer reaction time-of-flight mass spectrometer (PTR-MS, $[H_3O^+]$ reagent ion, Ionicon Analytik GmbH) (Bruns et
al., 2017), following the same procedure as in Klein et al. (2016). Additionally, elemental carbon (EC) mass
concentration was measured offline using a sunset thermo-optical analyzer, following the EUSAAR2 protocol
(Cavalli et al., 2010).
**2.2 Optical measurements**
**Aethalometer.** A dual-spot aethalometer (Magee Scientific aethalometer AE33, Aerosol d.o.o.) was used for real-
time aerosol light attenuation measurements at seven wavelengths ($\lambda$ = 370, 470, 520, 590, 660, 880 and 950 nm)
(Drinovec et al., 2015). The instrument measures the attenuation coefficient ($b_{ATN}$) of a light beam transmitted
through a filter tape loaded with aerosol samples. The use of the sampling flow (here, 2 L min$^{-1}$), integration time for
the measurement (here, 1 minute), and automated dual-spot loading compensation to obtain $b_{ATN}$ has been described
by Drinovec et al. (2015).
The loading compensated $b_{ATN}$ was used to infer the aerosol absorption coefficient, $b_{abs}$, using a constant wavelength
independent correction factor $C$, which accounts for multiple scattering within the filter matrix (Weingartner et al.,

117 2003):

$$b_{abs}(\lambda) = b_{ATN}(\lambda)/C \qquad (1)$$
As discussed in detail by Corbin et al. (2018), the wavelength-dependence of C can be expected to be negligible.
The loading compensated $b_{ATN}$ at 880 nm from the AE33 is further used to infer the equivalent-BC mass
concentration, $M_{eBC}$:
$$M_{eBC} = \frac{b_{ATN}(880\ nm)}{\sigma_{ATN}(880\ nm)} \qquad (2)$$
where $\sigma_{ATN}$ is the mass attenuation cross-section of BC deposited on the filter of the AE33. $M_{eBC}$ inferred from Eq.
(2) only equals the true BC mass concentration, $M_{BC}$, if the applied $\sigma_{ATN}$ is identical to the true attenuation cross-
section of BC, $\sigma_{\text{ATN,BC}}$, and if light attenuation at 880 nm is exclusively due to BC. $\sigma_{\text{ATN,BC}}(880\ nm)$ can be
inferred from the true MAC of BC, $\text{MAC}_{\text{BC}}$, and the true $C$ value:
$\sigma_{\text{ATN,BC}}(880\ nm) = \text{MAC}_{\text{BC}}(880\ nm) * C$  (3)
with $\text{MAC}_{\text{BC}}$ being defined as:
$\text{MAC}_{\text{BC}}(\lambda) = \dfrac{b_{\text{abs,BC}}(\lambda)}{M_{\text{BC}}}$  (4)
where $b_{\text{abs,BC}}$ is the absorption coefficient due to BC.
The manufacturer default values are 1.57 for $C$ (TFE-coated glass fiber filters) and 12.2 m$^2$ g$^{-1}$ for $\sigma_{\text{ATN}}$ at 880 nm,
which corresponds to a $\text{MAC}_{\text{BC}}$(880 nm) of 7.77 m$^2$ g$^{-1}$ (Gundel et al., 1984 , Drinovec et al., 2015). However, these
three parameters depend on aerosol properties. Here, we have determined the $C$ value by applying Eq. (1) to $b_{\text{ATN}}$
measured by the aethalometer and the absorption coefficient, $b_{\text{abs}_{\text{MWAA}}}$, measured by a multi-wavelength
absorbance analyser, MWAA (Massabò et al., 2015; Massabò et al., 2013). The $\text{MAC}_{\text{BC}}$(880 nm) was determined
using Eq. (4) to compare $b_{\text{abs}_{\text{MWAA}}}$ from the MWAA measurements with EC mass from the Sunset thermo-optical
analyzer (see Fig. 1A&B and Section 4.1 for detailed discussion). Following this procedure, the MWAA and Sunset
analyser will be defined as reference methods for absorption coefficient and EC mass concentration, respectively.
Note that data from these reference methods were only available with low time resolution and for a subset of all
samples. Thus, the aethalometer anchored against these reference methods, was used to obtain the wavelength
dependent absorption coefficients and the eBC mass concentrations with high time resolution using Eq. (1) and (2),
respectively. Processing the loading compensated AE33 attenuation coefficients with $C$ value and $\text{MAC}_{\text{BC}}$,
determined with independent MWAA and Sunset analyser measurements, ensures that the inferred $b_{\text{abs}}(\lambda)$ (Eq. (1))
and $M_{\text{eBC}}$ (Eq. (2)) have minimal bias compared to respective true values.
**MWAA measurements.** The MWAA (Massabò et al., 2015; Massabò et al., 2013) was used as reference method
for the aerosol absorption coefficient. It measures the absorption coefficient $b_{\text{abs}_{\text{MWAA}}}(\lambda)$ of particles deposited on
on standard filter samples. It is composed of five laser diodes, with $\lambda$ = 375, 407, 532, 635 and 850 nm, acting as
light sources and placed above the filter, an automated sample-changer, and three low-noise UV-enhanced
photodiodes. The first photodiode is placed behind the filter for transmittance measurements (0° relative to the
incident light, 1.5 cm from the sample), while the other two photodiodes are positioned at 125° and 165° (11 cm
from the sample) to collect the back scattered light. These transmittance and reflectance measurements are used
together with a radiative transfer model (Hänel et al., 1987) , which takes into account multiple scattering within the
particle/filter layer, to retrieve both the total optical thickness and the particle-filter-layer single scattering albedo,
providing the absorption coefficient $b_{\mathrm{abs_{MWAA}}}(\lambda)$ values. These calculations largely follow the approach
implemented in the multi-angle absorption photometer (MAAP, Petzold and Schönlinner, 2004).
**UV-visible absorbance measurements of extracted aerosols.** Filter samples were extracted for UV-visible
absorbance measurements in 10 mL ultrapure water or methanol in an ultrasonic bath for 20 min at 30 °C. Samples
were subsequently briefly vortexed (1 min) and filtered with 0.45 µm nylon membrane syringe filters following the
procedure described in Daellenbach et al. (2016). Absorption spectra were measured from 280 to 500 nm using a
UV-visible spectrophotometer (Ocean Optics) coupled to a 50-cm long-path detection cell (Krapf et al., 2016). Light
attenuation by the OA in solution, $ATN_{\mathrm{OA\text{-}sol}}$, at a given wavelength was recorded as the logarithm of the ratio of
signal intensities of the reference (solvent) ($I_0$) and the sample ($I$), both corrected for background signals with the
light source off.  From $ATN_{\mathrm{OA\text{-}sol}}$, the absorption coefficient of OA in solution, $b_{\mathrm{abs,OA-sol}}(\lambda)$, can be quantified as:
$$b_{\mathrm{abs,OA-sol}}(\lambda) = \frac{ATN_{\mathrm{OA-sol}}(\lambda)}{l} \qquad\qquad\qquad (5)$$
where $l$ is the optical path length.
The absorbance measurements are aimed at inferring the imaginary part of the refractive index. For this,
$b_{\mathrm{abs,OA-sol}}(\lambda)$ is transformed to the absorption coefficient of the bulk OA in the pure form, $b_{\mathrm{abs,OA-bulk}}$ (Sun et al.,

168     2007):

$$b_{\mathrm{abs,OA-bulk}}(\lambda) = b_{\mathrm{abs,OA-sol}}(\lambda) \left. \rho_{\mathrm{OA}} \middle/ \frac{m_{\mathrm{OA}}}{V_{\mathrm{solvent}}} \right. \qquad\qquad\qquad (6)$$
where $\rho_{\mathrm{OA}}$ is the bulk density of OA (assumed to be 1.5 g cm$^{-3}$, typical of wood-burning OA; (Corbin et al., 2015a;
Moosmüller et al., 2009; Sun et al., 2007)), $m_{\mathrm{OA}}$ is the extracted OA mass, and $V_{\mathrm{solvent}}$ is the solvent volume. The
bulk absorption coefficient directly leads to the imaginary part of the OA refractive index, $k_{\mathrm{OA}}$, in pure form
(Moosmüller et al., 2009):
$$k_{\mathrm{OA}}(\lambda) = b_{\mathrm{abs,OA-bulk}}(\lambda) \frac{\lambda}{4\pi} \tag{7}$$
Inserting Eq. (6) into Eq. (7) eventually provides (Liu et al., 2015a):
$$k_{\mathrm{OA}}(\lambda) = \frac{\lambda \rho_{\mathrm{OA}} V_{\mathrm{solvent}}}{4\pi m_{\mathrm{OA}}} b_{\mathrm{abs,OA-sol}}(\lambda) \tag{8}$$
The mass of organics dissolved in the solution could not be quantified. Therefore, we use an upper limit value for
$m_{\mathrm{OA}}$, approximated as the integral of AMS-measured OA mass concentration times sample flow rate over the filter-
sampling period. Accordingly, the resulting $k_{\mathrm{OA}}$ values represent lower limits for the true values, as the OA
extraction efficiency was not accounted for. If the OA extraction efficiency was less than unity, then the absorption
(or MAC) predicted from our solvent-extraction measurements would be less than that measured (or calculated)
using our real-time measurements (MWAA-calibrated aethalometer).
**2.3 Uncertainty analysis**
It is important to draw a clear distinction between uncertainties related to measurement precision and accuracy and
those related with experimental variability. In this section we discuss the quantifiable and unquantifiable
uncertainties related with the different measurements. In the result section, we will present our confidence levels on
the average parameters determined based on the experimental variability, which we judge to be the main source of
variance in the data.
**Quantifiable uncertainties.** The estimated uncertainty in the AMS-derived OA mass concentrations is ~25%,
which includes both potential biases and precision. This estimate is based on the variation in the AMS calibration
factors and estimated uncertainties in the SMPS used for the AMS calibration (Bruns et al., 2015, 2016).
Uncertainties related to particle transmission efficiency in the AMS are considered negligible for the particles
sampled here (Liu et al., 2007), whose volume size distribution falls within the range transmitted efficiently by the
AMS aerodynamic lens (see Fig. S4). The bounce-related collection efficiency (CE) of the AMS was concluded to
be unity for wood-burning OA in the literature reviewed by Corbin et al. (2015b; in their Section S1.2). For the
present data, the comparison between the SMPS mass (predicted from fitted volume distributions using a density of
1.5 g cm$^{-3}$) and the total PM predicted as AMS-OA+eBC, suggest a CE value between 0.7 and 1.0 (19% relative
uncertainty), consistent with average literature values and the uncertainties estimates. The uncertainty in EC mass
concentration, estimated from measurement repeats based on the EUSAAR2 protocol only, is within 7% in our case.

The precision uncertainty in the aethalometer attenuation measurements was estimated as 15 Mm$^{-1}$ based on the standard deviation of its signals prior to aerosol being injected into the smog chamber. The MWAA data have an estimated noise level and precision of 12 /Mm and 10% respectively, and these uncertainties have been added in quadrature to provide the overall uncertainties shown, for example, as error bars in Fig. 1 below. To compare the MWAA and aethalometer measurements, we determined $b_{abs,MWAA,880nm}$ by extrapolating the absorption coefficients measured at 850 nm to 880 nm using an α-value determined from the ratio between the absorption coefficients at 850 nm and 635nm. The uncertainty associated with this extrapolation is considered negligible relative to the overall MWAA uncertainty.

**Possible unquantified uncertainties.** There are significant uncertainties in the measurement of aerosol absorption using filter-based techniques (e.g., Collaud Coen et al., 2010). Here, we have used MWAA measurements as a reference to scale the aethalometer data, using a single C value. The correction factor C, which accounts for scattering effects within the filter matrix (Drinovec et al., 2015), may depend on the aerosol sample (Collaud Coen et al., 2010). In this study, we evaluated the variability in this factor for our primary and aged samples, by directly comparing the aethalometer to MWAA measurements, as discussed below. The MWAA has been previously validated against a polar nephelometer and a MAAP (Massabo et al., 2013), which, in turn, has been validated against numerous in situ methods (e.g., Slowik et al., 2007). The excellent correlation between MWAA and EC in our study (discussed below) supports the high confidence in the MWAA filter based absorption measurements conducted here. Another significant source of uncertainty in filter-based absorption measurements is the possible sorption (or evaporation) of volatile organics on (or from) the filter material. This may lead to an overestimation (or underestimation) of OA absorption. However, we have minimized sorption artefacts by utilizing a charcoal denuder. We have obtained an excellent correlation between OA absorption measurements derived from the MWAA-calibrated aethalometer and from quartz filter samples (see discussion below, Fig. 6 in the main text and S13 in the supplementary information). Although both of these techniques involved filter sampling, their sampling timescale is an order of magnitude different, and a difference is therefore expected if sorption (or evaporation) caused a substantial bias in our results. We therefore conclude that it is unlikely that artifacts associated with filter sampling have biased the absorption measurements. Finally, uncertainties related to pyrolysis during thermo-optical analysis may bias EC measurements. Such uncertainties arise from unstable organic compounds, and can be significant for biomass-burning samples, leading to biases on the order of 20% for EC (e.g. Schauer et al., 2003; Yang and Yu., 2007). To minimize these biases we applied the EUSAAR2 protocol. The optical properties of such organics are

generally different from BC; therefore, the excellent correlation between MWAA and EC data in Fig. 1A suggest
that pyrolysis effects were not a major source of uncertainty in our data set.

**3. OPTICAL PROPERTIES ANALYSIS**
**3.1 Determination of absorption Ångström exponents and mass absorption cross-sections**
In this section we describe the methodology adapted for the determination of the mass absorption cross-sections
(MACs) for the different aerosol material from the Sunset, MWAA and aethalometer measurements. The
assumptions and limitations underlying these calculations are clearly stated. We also explain the relationship
between the MACs and the wavelength dependence of the overall absorption.
**Definition of the absorption Ångström exponent .** The wavelength dependence of the overall absorption due to
both BC and BrC has often been described assuming a power law:
$b_{abs}(\lambda) \propto \lambda^{-\alpha}$ (9)
where α is the Ångström absorption exponent, often determined by fitting the absorption coefficient measurements
across the entire wavelength range. Eq. (9) is an empirical simplification, which breaks down when different
components having different spectral dependence contribute to the absorption, e.g. a mix of BrC and black carbon
(e.g., Moosmüller et al., 2011). In practice, different values of α would be obtained for different choices of $\lambda$ ranges,
and therefore we alternatively calculated two-wavelength absorption exponents according to
$\alpha(\lambda, \lambda_{ref}) = -\dfrac{\ln\left(\dfrac{b_{abs}(\lambda)}{b_{abs}(\lambda_{ref})}\right)}{\ln\left(\dfrac{\lambda}{\lambda_{ref}}\right)}$ (10)
where $\lambda$ is a wavelength of interest (in nm) and $\lambda_{ref}$ is the reference wavelength, here 880 nm. This reference
wavelength was chosen, because BC is expected to fully dominate light absorption in this range (Laskin et al.,

249 2015).

Black carbon is known to have an α between 0.9 and 1.1 (Bond et al., 2013; Kirchstetter et al., 2004; Liu et al.,
2015b), whereas BrC, which preferentially absorbs at shorter wavelength, has a higher α (Laskin et al., 2015; Saleh
et al., 2013). Thus, we interpret an increase of $\alpha(\lambda, \lambda_{ref})$ of the total aerosol as due to an increased contribution of
BrC to the total absorption. $\alpha(\lambda, \lambda_{ref})$ can potentially change due to other effects such as a wavelength dependent
lensing effect on absorption by BC (e.g., Lack and Langridge, 2013) or the restructuring of BC aggregates during
aging. The former effect was negligible under our conditions, as elaborated on below. The latter, if it occurs during
aging, would be attributed to SOA absorption in our approach. However, this is not an issue if our values are
accordingly applied in e.g. model simulations, following the same assumption as in our approach. This means that
the potential restructuring effects must implicitly be considered within the $MAC(\lambda)$ of SOA, while the $MAC(\lambda)$ of
BC must be kept fixed.
**3.2 Determination of $MAC_{BC}$ and $MAC_{POA}$ using the absorption Ångström exponent**
In a mixture of $n$ absorbing species, the total absorption at any wavelength may be written as the sum of the
absorbance of each of the species. Accordingly, Eq. (10) can be expressed for a multi-component system
$$\alpha(\lambda, \lambda_{\text{ref}}) = \frac{1}{\ln(\lambda_{ref}/\lambda)} \ln\left(\frac{\sum_{i=1}^{n} b_{\text{abs,i}}(\lambda)}{\sum_{i=1}^{n} b_{\text{abs,i}}(\lambda_{ref})}\right) = \frac{1}{\ln(\lambda_{ref}/\lambda)} \ln\left(\frac{\sum_{i=1}^{n} M_i MAC_i(\lambda)}{\sum_{i=1}^{n} M_i MAC_i(\lambda_{ref})}\right) \qquad (11)$$
where the right hand side follows the general definition of MAC along the lines of Eq. (4). $M_i$ and $MAC_i$ are the
mass concentration and MAC, respectively, of the $i^{\text{th}}$ species, with $n$ absorbing species in total. By considering that
the light absorption at $\lambda_{ref}$ = 880 nm is exclusively due to BC, and by defining BC to be the $n^{\text{th}}$ species, Eq. (11)
can be written as
$$\alpha(\lambda, 880nm) = \frac{1}{ln(880nm/\lambda)} ln\left(\frac{MAC_{BC}(\lambda)}{MAC_{BC}(880nm)} + \sum_{i=1}^{n-1} \frac{M_i MAC_i(\lambda)}{b_{\text{abs}}(880nm)}\right) \qquad (12)$$
In Eq. (12), the summation now only goes over the n-1 organic species, which contribute to light absorption.
The fresh combustion aerosol exclusively contains BC and POA as absorbing species. For the data at time $t_0$ before
the start of photo-oxidative aging, Eq. (12) simplifies to:
$$\alpha(t_0, \lambda, 880nm) = \alpha_{\text{BC+POA}}(t_0, \lambda, 880nm)$$
$\qquad = \frac{1}{\ln(880\text{nm}/\lambda)} \ln\left(\frac{\text{MAC}_{\text{BC}}(t_0,\lambda)}{\text{MAC}_{\text{BC}}(t_0,880nm)} + \frac{M_{\text{OA}}(t_0)\text{MAC}_{\text{POA}}(t_0,\lambda)}{b_{\text{abs}}(t_0,880nm)}\right)$ (13)
In Eq. (13), $M_{\text{OA}}(t_0)$ is the mass concentration of primary organic aerosol measured by the AMS at $t_0$.
$\text{MAC}_{\text{BC}}(t_0,880\text{nm})$ was inferred from the MWAA and Sunset thermo-optical analysis and shown to be independent
of the experimental conditions (Section 4.1; Fig. 1A). Absorption coefficients $b_{\text{abs}}(t_0,\lambda)$ are obtained from the high
time resolution attenuation measurements by the aethalometer referenced to the MWAA absorption measurements
as described above. $\alpha(t_0,\lambda,880\text{ nm})$ is derived from $b_{\text{abs}}(t_0,\lambda)$ and $b_{\text{abs}}(t_0,880\text{ nm})$ using Eq. (10). We have
intentionally formulated of Eq. (13) as such to highlight that the retrieved $\text{MAC}_{\text{OA}}(t,\lambda)$ depends mainly on the input
$M_{\text{OA}}$. Correspondingly, the retrieved $\text{MAC}_{\text{OA}}(t,\lambda)$ is mainly sensitive to potential AMS calibration biases. This leaves
only 2 free parameters in Eq. (13), $\text{MAC}_{\text{BC}}(t_0, \lambda)$ and $\text{MAC}_{\text{POA}}(t_0, \lambda)$. These were determined by fitting Eq. (13) to
$\alpha(t_0,\lambda,880\text{ nm})$, $M_{\text{OA}}(t_0)$, $\text{MAC}_{\text{BC}}(t_0,880\text{nm})$ and $b_{\text{abs}}(t_0,880nm)$ data measured in all experiments for fresh
emissions at $t_0$. This approach contains the implicit assumption that the two MAC values are also independent of
experimental conditions, and therefore these MACs should be considered as average values. The accuracy of these
MAC values obviously depends on the accuracy of the absorption and mass measurements. First, a systematic bias
in the $C$ value potentially caused by a systematic bias in the MWAA measurements propagates to an identical bias in
both $\text{MAC}_{\text{BC}}(t_0, \lambda)$ and $\text{MAC}_{\text{POA}}(t_0, \lambda)$. Second, a systematic bias in the Sunset EC mass measurements yields a
corresponding inverse bias in $\text{MAC}_{\text{BC}}(t_0, \lambda)$, while $\text{MAC}_{\text{POA}}(t_0, \lambda)$ remains unaffected. Third, a systematic bias in the
AMS POA mass yields a corresponding inverse bias in $\text{MAC}_{\text{POA}}(t_0, \lambda)$, while $\text{MAC}_{\text{BC}}(t_0, \lambda)$ remains unaffected. Eq.
(13) shows that $\alpha$ of the primary aerosol at a certain wavelength is largely driven by $\text{MAC}_{\text{POA}}(t_0,\lambda)$, i.e. the optical
properties of POA, and by the ratio $\frac{M_{\text{OA}}(t_0)}{b_{\text{abs}}(t_0,880nm)}$, which reflects the relative contributions of POA and BC to total
primary aerosol mass.
**3.3 Determination of MAC$_{\text{SOA}}$**
The MAC of SOA, MAC$_{\text{SOA}}$, can be generally defined as:
$\text{MAC}_{\text{SOA}} = \frac{b_{\text{abs,SOA}}}{M_{\text{SOA}}}$ (14)
where $b_{\text{abs,SOA}}$ and $M_{\text{SOA}}$ are the absorption coefficient and mass concentration of SOA, respectively. In the aged
aerosol, which contains the absorbing species BC, POA and SOA, $b_{\text{abs,SOA}}$ is the difference of the total absorption
minus the absorption by POA and BC:
$b_{\text{abs,SOA}}(t,\lambda) = b_{\text{abs}}(t,\lambda) - b_{\text{abs,POA+BC}}(t,\lambda)$        (15)
The absorption by POA and BC in the aged aerosol is a priori unknown, but can be calculated under certain
assumptions. The first assumption is that SOA does not contribute to absorption at 880 nm:
$b_{\text{abs,POA+BC}}(t, 880\ nm) \equiv b_{\text{abs}}(t, 880\ nm)$. The second assumption is that the two-$\lambda$ $\alpha$ values of primary emissions
do not change during aging $\alpha_{\text{POA+BC}}(t,\lambda, 880\ nm) \equiv \alpha_{\text{POA+BC}}(t_0,\lambda, 880\ nm)$. The latter approximation is based on
the underlying assumptions that the MAC of POA is not altered by aging and that the proportions of POA and BC
mass lost to the wall are identical. Under these assumptions $b_{\text{abs,POA+BC}}$ becomes:
$b_{\text{abs,POA+BC}}(t,\lambda) = b_{\text{abs}}(t, 880\ nm)\left(\frac{880\ nm}{\lambda}\right)^{\alpha_{\text{POA+BC}}(t_0,\lambda,880\ nm)}$        (16)
Note that inferring $b_{\text{abs,POA+BC}}(t,\lambda)$ from $b_{\text{abs}}(t, 880\ nm)$ implicitly accounts for the decrease in the BC and POA
absorption due to wall losses.
$M_{\text{SOA}}$ was obtained as total organic minus POA mass concentration:
$M_{\text{SOA}}(t) = M_{\text{OA}}(t) - M_{\text{POA}}(t)$        (17)
The POA mass concentration in the aged aerosol can be inferred from the initial OA mass concentration in the fresh
emissions by accounting for the wall losses using Eq. (S1) and the wall loss time constant $\tau$ (see Section Wall loss
corrections in the SI):
$M_{\text{POA}}(t) = M_{\text{OA}}(t_0)\exp(\tau^{-1}t)$        (18)
Inserting Eq. (15) - (18) into Eq. (14) provides the final equation for inferring $\text{MAC}_{\text{SOA}}$.
$\text{MAC}_{\text{SOA}}(t,\lambda) = \dfrac{b_{\text{abs}}(t,\lambda) - b_{\text{abs}}(t, 880\ nm)\left(\frac{880\ nm}{\lambda}\right)^{\alpha_{\text{POA+BC}}(t_0,\lambda,880\ nm)}}{M_{\text{OA}}(t) - M_{\text{OA}}(t_0)\exp(\tau^{-1}t)}$        (19)

$\text{MAC}_{\text{SOA}}$ can be calculated for every data point in time and for all aethalometer wavelengths from 370 to 660 nm
($\text{MAC}_{\text{SOA}}$ defined to be zero at $\lambda \geq 880$ nm), as all quantities on the right hand side of Eq. (19) are available from
either the aethalometer or AMS measurements or are otherwise known. It can be seen from Eq. (19) that the mass
concentrations used to calculate $\text{MAC}_{\text{SOA}}$ solely originate from AMS data, thus being consistent with the calculation
of $\text{MAC}_{\text{POA}}$ (see above). Eq. (19) is based on the assumption that POA is "chemically inert", i.e. no chemically
induced changes of $M_{POA}$ and $MAC_{POA}$ occur. Such chemically induced changes of absorption coefficient by POA,
through a change of $M_{POA}$ or $MAC_{POA}$, if they occur, are assigned to the absorption by SOA, thus resulting in a
corresponding adjustment of the inferred $MAC_{SOA}$.
**3.4 Mie calculation to relate $k_{OA}$ with $MAC_{OA}$**
The imaginary part of the refractive index of an aerosol component is an intensive material property. However, the
MAC of such an aerosol component additionally depends on the size and morphology of the aerosol (except for the
Rayleigh regime). The online aerosol absorption measurements provide estimates for MAC values, while the UV-
visible absorbance measurements of filter extracts provide the imaginary part of the refractive index. We used Mie
calculations in order to compare the two quantities. The $k_{OA}(\lambda)$ obtained from the filter extracts is converted to a
$MAC_{OA,bulk}$ by assuming that all OA is present in homogeneous spherical particles with a diameter distribution
identical to the mobility diameter distribution measured by the SMPS. In this manner, $MAC_{OA,bulk}$ becomes equal to
the mass-weighted average (=volume-weighted average) of the diameter dependent MAC:
$$MAC_{OA,bulk}(\lambda, n_{OA}, k_{OA}, \rho_{OA}) = \frac{\sum_i N_i d_i^3 MAC_i^{Mie}(\lambda, n_{OA}, k_{OA}, \rho_{OA})}{\sum_i N_i d_i^3} \qquad (20)$$
Here, $N_i$ and $d_i$ are the number of particles and particle diameter, respectively, in the $i$th size bin, and $n_{OA}$ is the real
part of the refractive index of the OA (which is assumed to be $n_{OA} = 1.5$ typical for organic material; Lu et al.,
2015). The MAC of particles with diameter $d_i$, $MAC_i^{Mie}$, was calculated using the Mie Code by Peña and Pal (2009)
(incorporated into Igor Pro 6.3, WaveMetrics, OR, USA by Taylor et al., 2015). $MAC_i^{Mie}$ also depends on the density
of OA, for which we assume a value of $\rho_{OA} = 1.5$ g cm$^{-3}$ (see Section 2.2), as the volume specific absorption cross-
section obtained from Mie theory needs to be converted to a mass specific absorption cross-section. We note that as
we have used the same value of $\rho_{OA}$ in the calculation of both $MAC_i^{Mie}$ and $k_{OA}(\lambda)$, $MAC_{OA,bulk}$ becomes
independent of the assumed $\rho_{OA}$ value.
Assuming spherical particles and neglecting the presence of BC in these particles may seem inappropriate. However,
calculations considering BC and assuming core-shell morphology revealed (1) limited sensitivity of the resulting
$MAC_{OA}$ to this assumption and (2) a higher than measured lensing effect. Therefore, a substantial fraction of the OA
seems to be externally mixed and to dominate the measured size distribution (see also Section 4.1).

## 4. RESULTS AND DISCUSSION

### 4.1 Verification of $MAC_{BC}$ and $C$ value



We have independently determined the $MAC_{BC}$(880nm) and the aethalometer C values under our conditions, as
follows. We determined $MAC_{BC}$(880nm) from the regression between the absorption coefficients at 880 nm
obtained from the MWAA and the EC mass measured by the Sunset analyzer (Fig. 1A). The slope of this regression
may be used to estimate the $MAC_{BC}$(880nm), which we retrieved as 4.7 ± 0.3 $m^2g^{-1}$ by an uncertainty-weighted
linear least-squares fit . The corresponding intercept was not significantly different from zero (-3 ± 3 /Mm). Our
$MAC_{BC}$(880nm) is not statistically significantly different from the value recommended by Bond et al., (2006) for
externally-mixed BC (extrapolating their $MAC_{BC}$(550nm) to 880 nm by assuming α=1 provides $MAC_{BC}$(880nm)=
4.7 ± 0.7 $m^2 g^{-1}$). The strong correlation between $b_{abs,MWAA,880nm}$ and EC in Fig. 1A shows that $MAC_{BC}$(880nm) did
not vary with aging during our study (see also Fig. S2-a). It also indicates that measurement artefacts for both
instruments were negligible, as the fundamental differences between the two techniques mean that any artefacts are
unlikely to be similar between them (charring for EC vs. adsorption artefacts for MWAA). Our absorption
coefficient measurements also provide insights into particle mixing state in this study. Since a single MAC
adequately described our samples at all levels of aging (Fig. 1A and Fig. S2-a), in spite of a factor of 3.3 average
increase in the aerosol mass, our samples cannot be adequately described by a core-shell Mie model. Such a core-
shell model would predict an absorption enhancement by a factor of ~1.8 (Bond et al., 2006) for the observed OA
mass increase with aging, which was not observed in our case. This observation is also supported by the time
resolved attenuation measurements at 880 nm using the aethalometer (Fig. S3), suggesting that little (<10%) to no
increase in the attenuation coefficients upon SOA formation. We emphasize that this conclusion does not indicate
that no internal mixing occurred, but rather that the simplified concept of negligible mixing better describes our data
than the equally simplified concept of a core-shell description of coatings that completely envelop the central BC
core. This may be due to the complex morphology of internally-mixed BC, which has been previously observed for
wood burning particles (e.g., China et al., 2013; Liu et al., 2015; Liu et al., 2017). It may also be related to the fact
that OA and BC are emitted during separate phases of combustion. OA rich particles are emitted during the pre-
flaming pyrolysis stage of combustion, whereas most BC is emitted during flaming combustion (Corbin et al.,
2015a, 2015b; Haslett et al., 2018; Heringa et al., 2011). These two stages of combustion may coexist in different
regions of the stove, particularly during simulated real-world usage. As lensing effect was negligible in our case, we
have assumed that the aerosol optically behaves as an external mix between BC and BrC during Mie calculation (see
section 3.4). We note that while this assumption is important for estimating the BC absorption, the conclusion drawn
about the BrC absorption are not very sensitive to the assumed morphology.
We determined time-resolved wavelength-dependent absorption coefficients as follows. We used the aethalometer to
obtain filter attenuation coefficients with high time resolution, which were then calibrated to obtain absorption
coefficients by deriving the factor C (Eq. (1)) using the MWAA measurements of filter samples. C was obtained
from an uncertainty-weighted linear least-squares fit as $3.0 \pm 0.2$ (Fig. 1B); the intercept of the fit was not
significantly different from zero, within two standard deviations ($-17 \pm 14$). A very strong correlation could be
observed between MWAA and aethalometer (Fig. 1B), implying that C is independent of the type of the aerosol
sampled (see also Fig. S2-B). Therefore, we used a single C value to obtain time-resolved wavelength-dependent
absorption coefficients from the aethalometer attenuation measurements at the different wavelengths for primary and
aged aerosols.
Note that the manufacturer's default values, which were not applied in our case, are 1.57 for C (using TFE-coated
glass fiber filters) and 12.2 $m^2$ $g^{-1}$ for $\sigma_{ATN}$ at 880 nm (Gundel et al., 1984 , Drinovec et al., 2015). The C value
determined here is larger than the manufacturer-default value for the AE33, resulting in smaller absorption
coefficients. However, the calculated $\sigma_{ATN}$ at 880 nm (13.8 $m^2$ $g^{-1}$), which can be retrieved as the product of the C
value and $MAC_{BC}(880nm)$ (Eq. (3)), is similar to the factory-default $\sigma_{ATN}$. Therefore, our calibrated $M_{eBC}$, calculated
from the attenuation coefficients using $\sigma_{ATN}$ (Eq. (2)), are similar to the factory-default $M_{eBC}$. We note that $M_{eBC}$ has
not been used for $MAC_{OA}$ calculations, and is only used for the calculation of the mass fractions of BC and OA for
display purposes (Fig. 2, 3, 7 and 8).
**4.2 Optical properties of BC, POA, and SOA**
In this section we derive the wavelength dependent mass absorption cross-sections for BC, POA and SOA. In Fig. 2,
we display the evolution of $\alpha(370nm, 880nm)$ as a function of OH exposure. Fig. 3 shows the relationship between
$\alpha(\lambda, 880nm)$ and $f_{OA}$ for primary and aged aerosols.
**α of primary emissions.** The $\alpha(370nm, 880nm)$ values computed for the primary aerosol (OH exposure = 0
molecules $cm^{-3}$ h) ranged between 1.3 and 1.7 (Fig. S5), which is within the range reported previously for biomass-
burning emissions (Kirchstetter et al., 2004; Lewis et al., 2008; Zotter et al., 2016). The $\alpha(\lambda, 880nm)$ is slightly
higher than that of pure BC (~1.2; Bond et al., 2013; Zotter et al., 2017) for small $f_{POA}$, while increasing $f_{POA}$
corresponded to a distinct increase of $\alpha(\lambda, 880nm)$. This increase provides clear evidence for the contribution of
primary BrC to the absorption at lower wavelengths (shown explicitly in Eq. (13)). The $f_{POA}$ ranges from 0.12 to
0.63, which is lower than $f_{POA}$ reported for open burning emissions (e.g., $f_{POA}$~0.75, Ulevicius et al (2016)), because
our wood-stove emissions feature a more efficient combustion. As illustrated in Fig. S5, the observed absorption
spectra have steeper gradients with decreasing wavelength compared to the lines of constant $\alpha$. Such systematic
increase in $\alpha(\lambda, 880nm)$ with decreasing $\lambda$ reflects the more-efficient light absorption by BrC at shorter wavelengths
(Moosmüller et al., 2011), and shows that the power law wavelength dependence is an inaccurate oversimplification
for this mixed aerosol.
**Evolution of α with aging.** Fig. 3B shows that upon aging, the OA fraction rapidly increased (a typical time series
of raw data is shown in Fig. S1), reaching an average value of 0.81 (full range for aged OA: $0.74 < f_{OA} < 0.89$) at
high OH exposures ($> 2{\times}10^7$ molecules cm$^{-3}$ h), and resulting in a corresponding increase of
$\alpha_{BC+POA+SOA}(370\text{nm}, 880\text{nm})$. The increase of $\alpha_{BC+POA+SOA}(370\text{nm}, 880\text{nm})$ and $f_{OA}$ were always correlated
and plateaued at OH exposures beyond ~$2{\times}10^7$ molecules cm$^{-3}$ h, as seen in Fig 2. Also, note in Fig. 2 that at highest
OH exposures, the highest $\alpha_{BC+POA+SOA}(370\text{nm}, 880\text{nm})$ were reached, on average 1.8, during experiments where the
$f_{OA}$ was highest. Such strong correlation between SOA formation and $\alpha_{BC+POA+SOA}(370\text{nm}, 880\text{nm})$ suggests the
production of substantial amounts of brown SOA. A similar relationship is observed between
$\alpha_{BC+POA+SOA}(\lambda, 880\text{nm})$ and $f_{OA}$ for higher wavelengths as shown in Fig. S6. Similar to the case of POA, a
systematic decrease in $\alpha(\lambda, 880nm)$ with increasing $\lambda$ is observed, reflecting the preferential absorption of BrC
SOA at shorter wavelengths. We note that $\alpha_{BC+POA+SOA}(370nm, 880nm)$ as a function of $f_{OA}$ for all experiments
lies below the overall trend for the primary aerosol (dashed line in Fig. 3B), implying that $MAC_{SOA}(370\text{nm})$ was
smaller than $MAC_{POA}(370\text{nm})$.
**Determination of MAC$_{BC}$ and MAC$_{POA}$.** We determined best-fit values for $MAC_{BC}(\lambda)$ and $MAC_{POA}(\lambda)$ from the
data shown in Fig. 3A. Fig. 3A includes least-squares fits of Eq. (13) to the data, with $MAC_{BC}(\lambda)$ and $MAC_{POA}(\lambda)$ as
fit parameters. The fit results are shown in Table 1. The obtained fit value of $MAC_{BC}(370\text{nm})$ was 13.7 m$^2$ g$^{-1}$ (GSD
1.1, one-sigma uncertainty 12.4—15.1 m$^2$/g), higher but not statistically significantly different from the range
estimated based on Bond et al. (2013), considering the uncertainties on both the $\alpha_{BC}$ values and the $MAC_{BC}(520\text{nm})$.
Meanwhile, the mean $MAC_{POA}$(370nm) value, equal to 5.5 m$^2$ g$^{-1}$, obtained under our conditions for domestic wood
burning is ~2.4 times higher than that obtained by Saleh et al. (2014) for open biomass burning primary emissions,
suggesting the presence of more-strongly absorbing organic material under our conditions (this comparison is
continued in Section 4.3).
**Determination of $MAC_{SOA}$.** The $MAC_{SOA}(\lambda)$ values, determined using Eq. (19), are shown in Fig. 4 and Table 1.
$MAC_{SOA}$(370nm) was 2.2 m$^2$ g$^{-1}$ (GSD 1.39), a factor of 2.5 smaller than $MAC_{POA}$(370nm), but approximately an
order of magnitude higher than values reported for ambient oxygenated aerosols or laboratory SOA from biogenic
and traditional anthropogenic precursors such as terpenes and methyl-benzenes (Clarke et al., 2007; Lambe et al.,
2013; Liu et al., 2016; Romonosky et al., 2015). The predominant SOA precursors identified in wood smoke
comprise (methyl)naphthalene(s) and phenol derivatives from lignin pyrolysis (Bruns et al., 2016; Ciarelli et al.,
2016), the oxidation products of which are expected to be highly light absorbing due to the presence of aromatic
moieties in the SOA (Bruns et al., 2016; Laskin et al., 2015). In this regard, it is not surprising that the
$MAC_{SOA}$(370nm) values obtained here are similarly high as those obtained from methanol-extracted SOA from
guaiacol and naphthalene oxidation (0.5–3.0 m$^2$ g$^{-1}$, Romonosky et al., 2015).
**Uncertainties and variability in $MAC_{BC}$, $MAC_{POA}$ and $MAC_{SOA}$.**
Table 1 shows the fitting errors related with $MAC_{BC}(\lambda)$, $MAC_{POA}(\lambda)$ and $MAC_{SOA}(\lambda)$, arising from our
measurement precision and experimental variability. These fitting errors are greater than our estimated uncertainties
in the absorption coefficients measured by MWAA (10%), and comparable to our estimated uncertainty in OA mass
measured by AMS (30%). The residuals in the fitted $MAC_{BC}(\lambda)$ are relatively low (< 10%), increasing with
decreasing $\lambda$. By contrast, the uncertainties in the fitted $MAC_{POA}(\lambda)$ are much higher (GSD = 1.2–1.5) and increase
with increasing $\lambda$. The relative residuals between the measured and fitted $\alpha(\lambda,880nm)$ for primary emissions showed
a mean bias and RMSE of 0.07 and 0.13, respectively (Fig. S7), indicating that our fitted MAC results provide a
good description of the data set. $MAC_{SOA}(\lambda)$ values determined were highly variable between experiments with a
GSD = 1.39 and 2.42 for $\lambda$=370 nm and 660 nm, respectively. In Fig. S10, we show the distribution of $MAC_{SOA}(\lambda)$
values as box and whiskers against OH exposure, showing no particular dependence of these values with aging as it
will be discussed below. Therefore, we expect the fitting errors in $MAC_{SOA}$ and of $MAC_{POA}$ to be mainly related to
true changes in the organic aerosol chemical composition between different burns, since the variability of $MAC_{BC}(\lambda)$
was relatively small. In Section 4.3, we discuss this variability further using the results of an additional and
independent analysis.
**MAC$_{BC}$, MAC$_{POA}$ and MAC$_{SOA}$ wavelength dependence.** The relationships between the MAC$_{SOA}(\lambda)$, MAC$_{POA}(\lambda)$
and MAC$_{BC}(\lambda)$ and wavelength appear to fall on three unique lines in the range 660 nm to 370 nm when plotted in
log-log space, as shown in Fig. 4 (Fig. S8 shows the same data plotted on a linear scale). This indicates that a power-
law approximation provides a good description of the behavior of individual components within this wavelength
range from 370 nm to 660 nm. Accordingly we fitted the power law coefficients to the data shown in Fig. 4
$(\ln(MAC_i) = \ln(A_i) + \alpha_i \ln(\lambda)$, with $i=$ BC, POA, or SOA) and fitting parameters are shown as multivariate
probability density functions in Fig. S9. This yielded $\alpha_{BC} = 1.2$, $\alpha_{POA} = 4.6$, and $\alpha_{SOA} = 5.6$, with corresponding
uncertainties of approximately 20% (complete details of the uncertainties are provided in Table S1). Note that $\alpha_{BC}$ in
the range 660 nm to 370 nm obtained from this fit is very similar to $\alpha_{BC}$ values that can be inferred by extrapolating
the data shown in Fig. 3A to $f_{OA}=0$. The high $\alpha$ values obtained for the organic fractions are consistent with previous
measurements for BrC containing POA (e.g. Chakrabarty et al., 2010, 2013).
**Evolution of MAC$_{OA}$ with aging.** In Fig. 5, we examine whether the absorption profile of SOA evolved with aging.
A change in MAC$_{SOA}(370nm)$ or $\alpha_{SOA}$ with increasing OH exposure may indicate either a change in the mass-
specific absorption of the condensing SOA species with time, or a change (e.g. "bleaching") in the MAC of pre-
existing POA. Fig. 5 indicates that neither of these scenarios was the case. Both MAC$_{SOA}(370nm)$ and $\alpha_{SOA}$ were
statistically independent of the OH exposure, for exposures up to 40 molec. OH cm$^{-3}$ h. This signifies that under our
conditions and within our measurement uncertainties the optical properties of the additional organic mass formed
was constant with aging, under the assumption that the light-absorption properties of POA were negligibly
influenced by aging. Most of the variability in MAC$_{SOA}(\lambda)$ discussed above is therefore related to experiment-to-
experiment differences rather than to the extent of OH exposure, as it is also shown below.
**4.3  Solubility of BrC in methanol and water**
Fig. 6 shows the MAC$_{OA}(370nm)$ determined from the water and methanol extracts against the MAC$_{OA}(370nm)$
determined from the online measurements. The MAC$_{OA}(370nm)$ from online measurements was estimated by
subtracting the contribution of BC assuming a constant MAC$_{BC}(370nm) = 13.7$ m$^2$.g$^{-1}$ as obtained in this work
(Table 1). We performed all the calculations and comparisons at $\lambda = 370$ nm, as the signal to noise ratio of the
absorption coefficients measured by UV-visible spectroscopy and the contribution of BrC to the total carbonaceous
absorption are highest at this wavelength. The MAC of the extracts was computed from the $k_{OA}$ through Mie
calculations. Repetition of both water and methanol extracts yielded results that were consistent within 10% (Fig.
S11). Average raw absorption spectra are shown in Fig. S12.
Fig. 6B shows excellent correlation between the $MAC_{OA}$(370nm) values obtained from the kOA of the solvent-
extracted OA with the in-situ method described above. The Pearson correlation coefficient was 0.8, for both
solvents. This correlation suggests that none of the assumptions employed in either method led to substantial errors
in precision, providing direct support for our results. A similar relationship was observed between $k_{OA}$ and the
$MAC_{OA}$(370nm) determined from the online measurements (Fig. S13), showing that this relationship is not sensitive
to assumptions underlying the Mie calculations.  It further suggests that the wide variability observed in the $MAC_{OA}$
values of different burns, as seen Fig. 6, most likely reflects real variability in the optical properties of POA and
SOA rather than random noise or experimental errors in the retrieved quantities. $MAC_{OA}$ retrieved based on the $k_{OA}$
of the water soluble OA show substantially more scatter than observed in Fig. 6B (for both primary and aged data),
suggesting a variable extraction efficiency in the case of water, which we also attribute to variability in the OA
composition.
The data in Fig. 6B show that the methanol extracts correspond to a MAC about 50% smaller than the online data.
The scatter in the data is significantly reduced for the aged data (note that, in this analysis, aged OA refers to the
sum of POA and SOA, since the reported values represent all OA after aging). This reduced scatter is expected,
considering that aging is likely to result in more-spherical particles.  We have assumed particle sphericity when
interpreting the SMPS data and performing the Mie analysis. While the propagation of quantifiable uncertainties
leads to an error estimate of ~25%, considering the simplifiations that were necessary for the Mie analysis, we
consider a 50% closure to be an adequate agreement. Despite this, we cannot exclude additional methanol insoluble
brown carbon. Conversely, the fit in Fig. 6A indicates that the apparent MAC of water-soluble species was a fourth
of the respective methanol MAC, according to the slope of only 12 ± 3%. Only the aged data have been fit to
illustrate this point. This strong disagreement shows that the BrC in our samples was hardly water soluble, even for
the most aged samples. As we expect that the majority of OA in our samples formed by wood pyrolysis (Di Blasi,
2008; Corbin et al., 2015b; Shafizadeh, 1984), we can compare our results directly to those of Chen and Bond
(2010), who also found that primary wood-pyrolysis BrC was water insoluble. Moreover, the poor water solubility
of the light absorbing components of SOA (Zhang et al., 2011) is in line with the results by Bruns et al. (2016) who
showed that SOA precursors during these experiments were predominantly aromatic compounds.
**4.4 Comparison of $k_{OA}$ with literature**
The results above highlight the variability in the OA absorption properties. In this section, we discuss potential
reasons for this variability and compare our results to literature. Fig. 7 shows the imaginary refractive index of
methanol-extracted OA at 370 nm, $k_{OA,methanol}(370nm)$ (Eq. (8)), as a function of $M_{BC}/M_{OA}$ and aging. The data are
plotted against $M_{BC}/M_{OA}$ instead of $f_{OA}$ to allow for a direct comparison with literature (see Fig. S14 for a plot
against $f_{OA}$). An approximately linear trend of $k_{OA,methanol}(370nm)$ with $M_{BC}/M_{OA}$ is seen in log space. This aging-
independent relationship may be useful in, for example, atmospheric scenarios where wood-burning OA is a
dominant aerosol component but its exact degree of aging is unknown. The decrease of $M_{BC}/M_{OA}$ caused by
formation of SOA during aging results in a concurrent decrease of $k_{OA,methanol}(370nm)$, implying that $k_{SOA} < k_{POA}$.
This result is consistent with the smaller MAC of SOA compared to POA obtained from online measurements
(Table 1) and with recent results reported by Sumlin et al. (2017) . We emphasize that the derived quantity here is
the imaginary refractive index $k$ of the total aged OA, not the SOA.
The increase of $k_{OA,methanol}(370nm)$ with increasing $M_{BC}/M_{OA}$ indicates that the OA compounds present at higher
$M_{BC}/M_{OA}$ absorbed more efficiently than at low $M_{BC}/M_{OA}$. If the variability in $M_{BC}/M_{OA}$ was driven partly by OA
partitioning, then this implies that lower-volatility compounds were more absorbing than high-volatility compounds,
consistent with the results by Saleh et al. (2014) who investigated the relation between OA absorption and volatility
using thermodesorber measurements. A correlation between $k_{OA}$ and $M_{BC}/M_{OA}$ has also been reported by Lu et al.
(2015). The parameterizations reported by these authors are included in Fig. 7, where the wavelength dependence
reported by those authors has been used to adjust their parameterizations to 370nm. Despite these differences, our
results confirm the generality of the correlation proposed by Saleh et al. (2014), but using a method that is
independent of potential biases related to internal mixing effects, filter-based absorption measurements or Mie
calculations. Indeed, we emphasize that the $k_{OA}$ obtained here is a lower limit: as our approach does not account for
the OA extraction efficiency; $k_{OA,methanol}(370nm)$ may be underestimated by up to a factor of ~2, based on Fig. 6B.



## 5. ATMOSPHERIC IMPLICATIONS


In this section, we seek to estimate the relative importance of OA absorption at different wavelengths relative to that
of the total carbonaceous aerosol as a function of aging. For these calculations, the MAC($\lambda$) values for the different
components and their relative mass abundance are required. We used the power law parameters reported above to
generate continuous $MAC_{BC}(\lambda)$, $MAC_{POA}(\lambda)$, and $MAC_{SOA}(\lambda)$ functions together with their associated uncertainties
(Fig. 8A), which allow the extrapolation of these parameters in the range [280nm; 880nm].
The contributions of the different components as a function of OH exposure were calculated by assuming that SOA
production follows the first order decay of its precursors, i.e., the reaction with OH. Under this assumption, the time-
dependent mass concentration of SOA compared to POA can be expressed as
$$M_{SOA,WLC}(t)/M_{POA,WLC}(t) = M_{SOAP,WLC}/M_{POA,WLC} \times \left(1 - \exp\left(-k_{OH}OH_{exp}\right)\right) \qquad (21)$$
In this equation, $M_{SOA,WLC}(t)$, $M_{POA,WLC}(t)$ and $M_{SOAP,WLC}$ are the wall loss corrected mass concentrations of SOA,
POA and the SOA potential (the maximum SOA formed upon the consumption of all precursors). $k_{OH}$ represents an
estimation of reaction rate of SOA precursors towards OH based on SOA production rates. By fitting the observed
$M_{SOA,WLC}(t)/M_{POA,WLC}(t)$ against the OH exposure, $k_{OH}$ and $M_{SOAP,WLC}/M_{POA,WLC}$ can be estimated. For these
calculations, we have estimated the wall losses using two approaches as described in the SI.
The $M_{SOAP,WLC}/M_{POA,WLC}$ was on average equal to 7.8 (GSD = 1.4) and $k$OH was estimated as $2.7\times10^{-11}$ molecule$^{-1}$
cm$^3$ (GSD = 1.4), consistent with the SOA precursors chemical nature measured (e.g. PAH and phenol derivatives)
by a proton-transfer-reaction mass spectrometer (PTR-MS) (Bruns et al., 2016, 2017). These high rates and
enhancement ratios indicate the rapid production of SOA.
Based on the bulk gas phase measurements of SOA precursors (Bruns et al., 2016), the obtained enhancements are
consistent with high bulk SOA yields of ~50%. These high values are not surprising, considering the nature of these
gases (e.g. PAH and phenol derivatives), the low temperatures (263 K), and the relatively high concentrations (Aged
OA ~100 µg m$^{-3}$) at which the experiments have been conducted (Bruns et al. 2016).
Combining these calculated enhancements with the average contributions of POA in primary emissions, the
evolution of $f_{OA}$ with aging was determined and is shown in Fig. 8B. The uncertainties in Fig. 8B (dotted lines)
represent one standard deviation on $f_{OA}$ obtained by a Monte Carlo propagation of uncertainties due to experiment-
to-experiment variability, fitting errors and wall loss correction errors (see SI). While this calculation represents a
simplification of the SOA production mechanisms (the dependence of SOA yields on OH exposures/multigeneration
chemistry and OA mass concentrations was neglected), it results in residuals much smaller than the experiment-to-
experiment variability. We therefore used these calculations to assess the relative contribution of OA to the total
carbonaceous absorption. We show in Fig. 8C that below 400 nm and upon aging, the absorption coefficient of the
total organics was at least as high as the one of BC.
Using the MAC values of the different components (in $m^2$ $g^{-1}$), their abundance (in g $m^{-3}$) and the solar irradiance
data (*S,* in W $m^{-2}$ $nm^{-1}$) calculated at sea level for a cloudless day, the fractional energy transfer due to the BrC light
absorption relative to that due to the total carbonaceous aerosol absorption , $W_{OA}(OH_{exp})$, in air masses dominated
by residential burning emissions can be determined as
$W_{OA}(OH_{exp}) = RET_{OA}(OH_{exp})/RET_{tot}(OH_{exp})$
$= \dfrac{\int_{300}^{880}\{M_{POA}(OH_{exp}) \times MAC_{POA}(\lambda) + M_{SOA}(OH_{exp}) \times MAC_{SOA}(\lambda)\} \times S(\lambda) \times d\lambda}{\int_{300}^{880}\{M_{BC}(OH_{exp}) \times MAC_{BC}(\lambda) + M_{POA}(OH_{exp}) \times MAC_{POA}(\lambda) + M_{SOA}(OH_{exp}) \times MAC_{SOA}(\lambda)\} \times S(\lambda) \times d\lambda}$    (22)
Here, $RET_{OA}(OH_{exp})$ and $RET_{tot}(OH_{exp})$ denote the rate of energy transfer per volume (in W $m^{-3}$) to the air mass in
question due to light absorption by OA and the total carbonaceous aerosol, respectively. We note that while
$RET_{OA}(OH_{exp})$ and $RET_{tot}(OH_{exp})$ are extensive properties, $W_{OA}(OH_{exp})$ does not depend on the loading or
scattering/lensing, provided that scattering/lensing similarly affects BC and OA present in the same air mass (e.g.
BC and OA have a similar size distribution).
We also note that $W_{OA}(OH_{exp})$ depends on the photon flux, *S(λ)*, but we consider this dependence to be trivial
compared to the variability in the aerosol emissions and their light absorbing properties (error bars considering these
variabilities are shown in Fig. 8D). Errors in $W_{OA}$ were propagated by Monte Carlo simulations using the
uncertainties from the estimated MAC values of BC and OA fractions and the variability in $f_{OA}$. Our sensitivity
analysis suggests that the major part of the variance in predicting $W_{OA}$ for primary emissions stems from the
variability in the POA mass fraction. In contrast, the SOA mass absorption cross-sections at lower wavelengths are
the most critical factor for assessing the relative importance of BrC absorptivity in aged emissions.
Fig. 8D shows that the fractional energy transfer to the air mass, $W_{OA}$, due to the absorption by the primary organic
aerosol was around 10% of that of the total carbonaceous aerosol for our samples. This percentage is comparable to
that observed by Fu et al. (2012), in spite of $f_{OA}$ in their samples being much higher, because of the high OA MACs
in our samples (Table 1). Moreover, with aging, the fraction of OA is enhanced, resulting in a sizeable increase $W_{OA}$,
from ~0.1 to ~0.3 (Fig. 8D), highlighting that SOA formation in biomass burning plumes is an atmospherically
relevant source of BrC. We note that our data are more representative of flaming conditions. More data are needed
on the chemical nature of primary particulate emissions and of the contributing SOA precursors as well as the
absorptivity of these primary and secondary products, for better constraining the influence of biomass-burning
related BrC on the Earth's climate.

**6. CONCLUSIONS**
We determined wavelength-dependent MAC values of BC, POA and SOA, as well as $k_{OA}$ for methanol and water
extracts of fresh and aged OA, for wood-burning emissions through smog-chamber experiments. To our knowledge,
this is the first determination of these properties for wood-burning OA. We showed that the $MAC_{OA}(370nm)$ values
calculated based on $k_{OA}$ through Mie analysis correlated well with those estimated from online filter based
measurements. This correlation between independent MAC measurements supports the quality of both methods.
While $MAC_{OA}(370nm)$ values computed based on $k_{OA,methanol}$ were a 2-fold lower than those estimated from online
filter based measurements, calculations based on $k_{OA,water}$ could only explain 12% of the measured absorption,
suggesting that BrC species in POA and SOA are mostly water insoluble. The $MAC_{OA}$ was found to vary by more
than one order of magnitude. Similar to previous reports, this variability could be related to the variability in the
ratio of the mass concentrations of BC and OA ($M_{BC}/M_{OA}$) due to very different mechanisms of oxidative aging and
burn-to-burn variability.
The $MAC_{POA}$ and $MAC_{SOA}$ determined for wavelengths between 370 and 660 nm followed a power-law dependence
on λ with an absorption Ångström exponent of 4.6 and 5.6 for POA and SOA, respectively. In addition to following
this power law, the MACs of POA and SOA appeared to be constant for OH exposures up to 40 x $10^6$ molecules cm$^-$
$^3$ h.
The mean $MAC_{POA}(370nm)$ obtained under our conditions was 5.5 m$^2$ g$^{-1}$, considerably higher than previously

reported values for open biomass burning. The mean $MAC_{SOA}$(370nm) was 2.2 m$^2$ g$^{-1}$ (one-sigma variability: 1.6 –

3.1 m$^2$ g$^{-1}$ according to a GSD = 1.39) under our experimental conditions, 2.3 times lower than the mean

$MAC_{POA}$(370nm) but approximately an order of magnitude higher than MAC values estimated for ambient

oxygenated aerosols or reported for SOA from biogenic and traditional anthropogenic precursors. We propose that

the important role of oxidized phenols and aromatics in forming wood-burning SOA (Bruns et al., 2016) is the cause

of this observation. This hypothesis is supported by our observed reaction rates with OH, and by the water-

insolubility of the BrC in aged OA.

Overall, the absorption by organic aerosols was estimated to contribute 10-30% of the total solar absorption of

wood-combustion aerosols, where 10% represents the primary OA and 30% the aged OA. SOA formation in

biomass burning plumes is therefore an atmospherically relevant source of BrC.

*Acknowledgements.* The research leading to these results has received funding from the European Research Council

grant (ERC-CoG 615922-BLACARAT) and by the Competence Centre Energy and Mobility (CCEM) project 807.

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

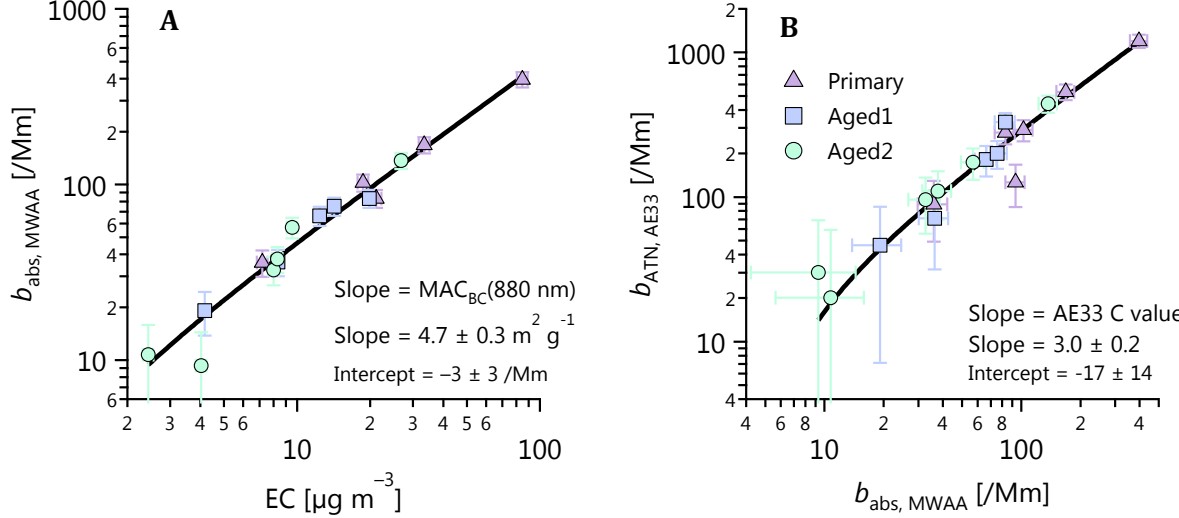


**Figure 1: Determination of (A) $MAC_{BC}$(880nm) and (B) aethalometer $C$ value using MWAA absorption measurements,**
**thermal/optical EC (EUSAAR2 protocol) and aethalometer attenuation measurements. MWAA absorption measurements**
**at 880 nm is determined by extrapolating the absorption coefficients at 850 nm using an $\alpha$ determined from the ratio**
**between the absorption coefficients at 850 nm and 635nm. The aerosols were either primary (no OH exposure), Aged 1**
**($\sim$1x10$^7$ molec OH cm$^{-3}$ h), or Aged 2 ($\sim$4x10$^7$ molec OH cm$^{-3}$ h). No difference in MAC or $C$ value was discernable with**
**aging (see also Fig. S2). The $C$ value derived from $\sigma_{ATN}$ recommended by Drinovec et al. (2015) = 2.6 compares well with**
**the value derived in Fig. 1B.**






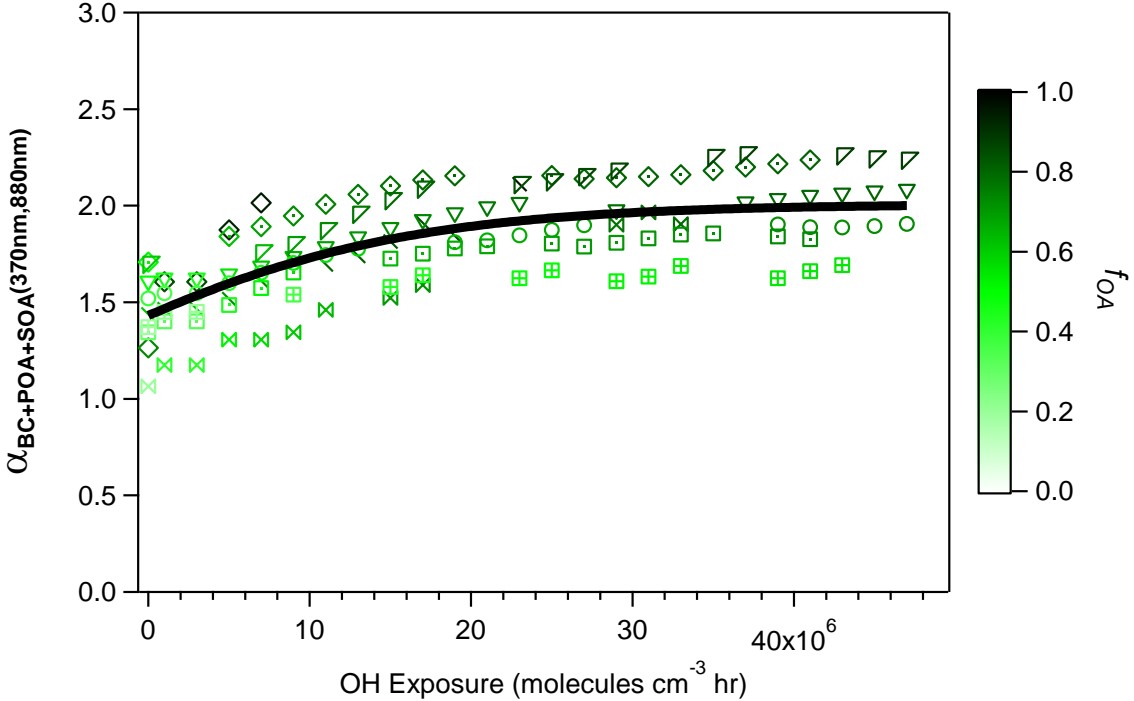


**Figure 2: Evolution during photochemical aging of $\alpha_{\text{BC+POA+SOA}}(370\text{nm}, 880\text{nm})$ (two-wavelength Ångström exponent**
**calculated using total absorption data at 370 nm and 880 nm), where the different symbols denote individual experiments.**
**Data are colored by the OA mass fraction $f_{\text{OA}} = M_{\text{OA}}/(M_{\text{OA}} + M_{\text{BC}})$. The black line is a fit to guide the eye.**

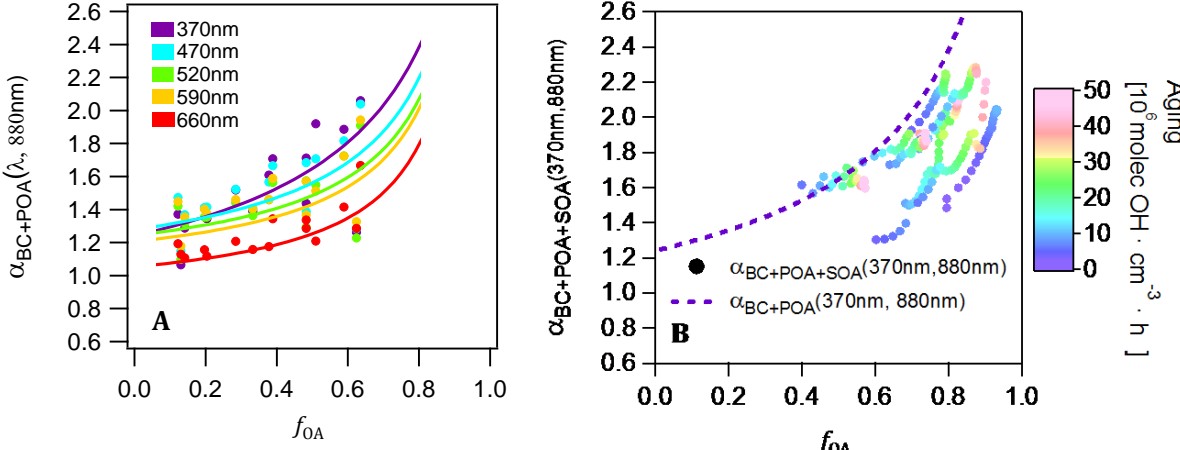


**Figure 3: (A) Relationship of $\alpha_{\text{BC+POA}}$ ($\lambda$, 880nm) to $f_{OA}$ for seven wavelengths for primary emissions. Data are colored**
**by the wavelength. Curves are fits of Eq. (13) to the data. Each point represents the average of one experiment and**
**therefore the variability in $f_{OA}$ is related to the variability in the emission composition between experiments. (B)**
**Relationship of $\alpha_{\text{BC+POA+SOA}}(370\text{nm}, 880\text{nm})$ to $f_{OA}$ for several experiments for aged aerosols. Data are color coded by**
**the OH exposure. The variability in $f_{OA}$ is due to SOA formation with aging; data from several experiments are shown**
**which explains the wide range of $f_{OA}$ at low OH exposures. Note that more data are included in A than B, as primary**
**emissions for some experiments were not aged**.


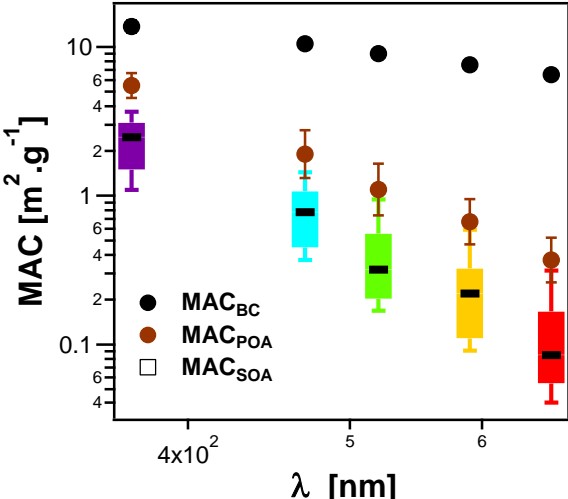


**Figure 4: MAC$_{SOA}$(λ) calculated from several smog chamber experiments plotted as box-whiskers as a function of wavelength (also shown by the color of the bars). The thick black lines, the boxes and the whiskers mark the medians, the quartiles and the 10th and the 90th percentiles, respectively. Also shown are the MAC$_{BC}$(λ) and MAC$_{POA}$(λ) reported in Table 1. Note that MAC$_{SOA}$(880nm) and MAC$_{POA}$(880nm) are zero by definition.**

941

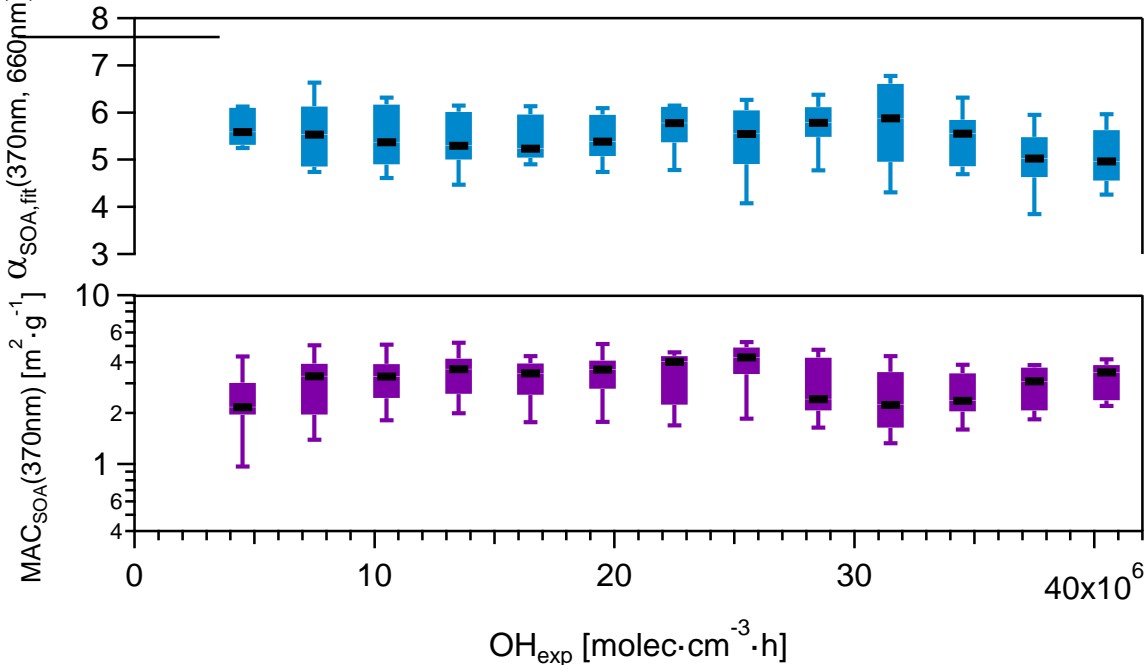

942

**Figure 5: MAC$_{SOA}$(370nm) and $\alpha_{SOA,fit}$(370nm, 660nm) calculated from several smog chamber experiments plotted as a function of OH exposure. The box marks the 25th and 75th percentile, while the whiskers mark the 10th and the 90th percentile. MAC$_{SOA}$(370nm) was obtained using Eq. (19). $\alpha_{SOA,fit}$(370nm, 660nm) was obtained from fitting the MAC$_{SOA}$ values in the range 370-660 nm for the different experiments against the wavelength. $\alpha_{SOA,fit}$(370nm, 660nm) is the slope of the linear fit applied after log transforming the data. MAC$_{SOA}$(λ) for higher wavelengths are shown in Fig. S10.**

948

949

950

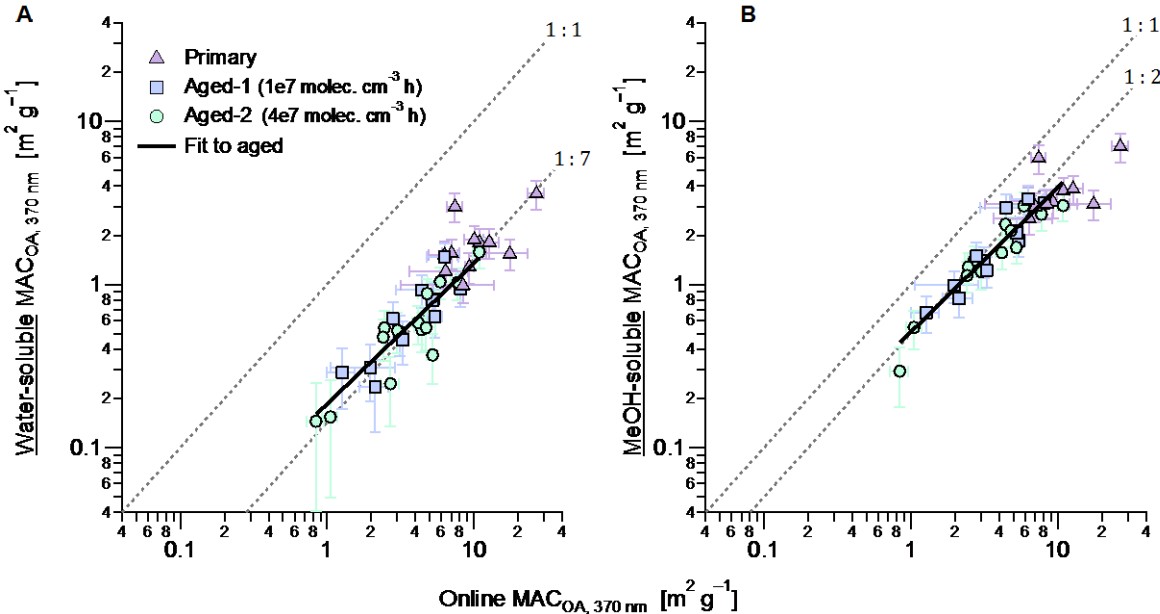

**Figure 6: Comparison of the $MAC_{OA}$(370nm) of aged aerosols determined from online and offline absorption measurements. The offline filter-extraction method directly quantified properties of total OA (ordinate), while the average of $MAC_{SOA}$ and $MAC_{POA}$ weighted with respective mass concentrations is shown on the abscissa. The panels show offline measurements of (A) water-soluble OA, (B) methanol-soluble OA. Fits are to aged data only due to the significantly smaller scatter of those data, although primary data on average follow similar trends. The fitted slopes and intercepts are, respectively, (A) $0.13 \pm 0.02$ and $0.05 \pm 0.06$ $m^2g^{-1}$ and (B) $0.12 \pm 0.1$ and $0.38 \pm 0.03$ $m^2g^{-1}$.**

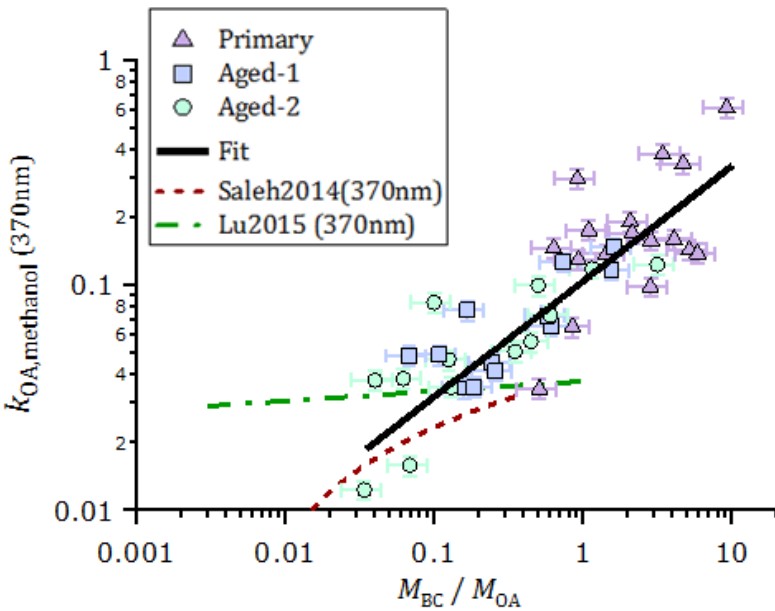

961

**Figure 7: Imaginary part of the OA refractive index at 370 nm, obtained from offline UV/vis spectroscopy of methanol OA extracts, plotted as a function of $f_{OA}$. The data could be empirically represented by a linear function in the log-log space, in the measurement range. The ordinary least-squares fit is $(k_{OA,nm}) = \log(M_{BC}/M_{OA})(0.51\pm0.07) + (-0.98\pm0.05)$. Also shown are parameterizations of $k_{OA}(370\ nm)$ for open burning against $M_{BC}/M_{OA}$ estimated based on the online $k_{OA}$ (550 nm) measurements in Saleh et al. (2014) and Lu et al. (2015), using the $k_{OA}$ wavelength dependence reported by the respective authors.**

968

969

970

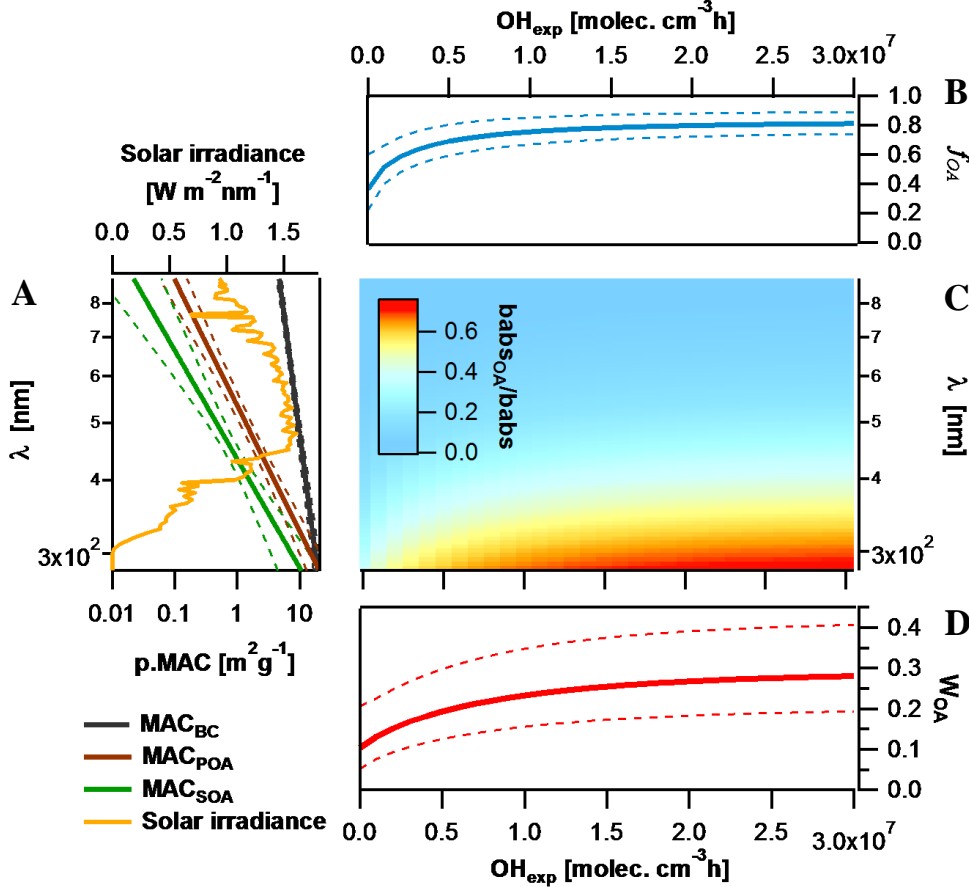

Figure 8: **Impact of BrC absorption on total primary and secondary wood-burning-aerosol absorption. (A) MACs of different particle components (BC, POA and SOA) along with their corresponding standard deviations plotted as a function of wavelength based on smog chamber data and compared to the solar irradiance spectrum. (B) Species average relative abundance in the smog chamber ($f_{OA}$) plotted as a function of the OH exposure. (C) Image plot showing the OA absorption coefficient relative to the total aerosol absorption as a function of wavelength and OH exposure. (D) Rate of energy transfer due to BrC light absorption relative to the total carbonaceous aerosol absorption ($W_{OA}$) estimated as a function of aging using the solar flux, the fractions of the different components and their MACs.**

990

**Table 1: Geometric mean and standard deviations of the determined MACs of BC, POA and SOA at different wavelengths. Uncertainties were obtained from fits of Eq. (13) for $MAC_{BC}$, $MAC_{POA}$, while for $MAC_{SOA}$ uncertainties GSD values are geometric standard deviation values on the $MAC_{SOA}$ average values from all experiments. These uncertainties do not include uncertainties related to the determination of $MAC_{BC}(880nm)$. By definition, BrC absorbance at 880 nm is zero.**

| | BC | | POA | | SOA | |
|---|---|---|---|---|---|---|
| $\lambda$ (nm) | GM ($m^2\ g^{-1}$) | GSD | GM ($m^2\ g^{-1}$) | GSD | GM ($m^2\ g^{-1}$) | GSD |
| 370 | 13.7 | 1.1 | 5.5 | 1.21 | 2.2 | 1.39 |
| 470 | 10.5 | 1.06 | 1.9 | 1.45 | 0.72 | 1.61 |
| 520 | 9 | 1.04 | 1.1 | 1.49 | 0.34 | 1.75 |
| 590 | 7.6 | 1.03 | 0.67 | 1.42 | 0.2 | 1.97 |
| 660 | 6.5 | 1.01 | 0.37 | 1.41 | 0.09 | 2.42 |
| 880 | 4.6 | 0.7 | 0* | | 0* | |

996       *By definition

997