# Peer review of "Production of particulate brown carbon during atmospheric"

_Atmospheric Chemistry and Physics, 2018_

## Referee Comment (RC1) · Anonymous Referee #1 · 31 May 2018

**GENERAL COMMENT**

The paper presents an analysis of black carbon (BC) and organic aerosol absorption properties upon aging. The experiments were done in a laboratory using a smog chamber were aerosols were exposed to OH radical and UV radiation. Several optical and chemical properties were measured online and filter samples were collected to be analyzed later offline by a multiple wavelength absorbance measurement technique and an EC/OC analyzer. Methanol and water filter extracts were also analyzed and absorbance was measured. The data is very valuable and the paper is well written and presented. I would recommend its publication after addressing the comments I present

below.

My main concerns are related to the following aspects:

- The offline techniques used in this study suffer of different artifacts and they are not sufficiently discussed in the manuscript.

- In this study the BC particles were not observed to be coated with other kind of particles (i.e., no internal mixing but external mixing was observed). However, ambient studies have shown that BC particles in the atmosphere are usually coated and this coating causes an enhancement of BC absorption. Given that, how representative is this study of "atmospheric aging"? Were the experiments not long enough to "age" the BC particles?

SPECIFIC COMMENTS

- Lines 107-108: How can you guarantee the correction factor C is wavelength independent?

- Line 122: "1.57 for C". Does this mean your Aethalometer was using TFE-coated glass fiber filters? Please mention the filter material.

- Line 128: It should be necessary to add some more discussion about possible artifacts that affect both techniques (MWAA and Sunset analyser).

- Line 136. MWAA measurements. Which artifacts are to be considered when using this technique in comparison to the Aethalometer? Can you provide more evidence on the comparison of this technique to other absorption measurements like MAAP or PAS?

- Line 171: What do you mean with the online kOA? How was this measured?

- Lines 403-411: How does this result compares to other studies? Ambient measurements have shown quick oxidation of brown carbon chromophores. Please comment about it.

TECHNICAL CORRECTIONS

- Line 101: It would be convenient to add numbers to these headlines across the manuscript for the sake of readability (e.g., "2.2.1 Aethalometer").

- Line 123: at?

- Line 141: Which angles?

- Lines 294-295: Please rephrase this sentence to improve understanding.

- Lines 296-299: It can be found awkward that the two variables needed to calculate MAC are coming from the same measurement technique (Aethalometer). Please try to sustain the reasons why it was done this way.

- Line 372: It should be written "Eq. (19)", and "Fig. 4". Please implement this across the manuscript. Check the journal guidelines.

- Lines 542-543: Please add uncertainty intervals to the reported MAC values.

- Figures 3A, S6, and S10: The data is presented using discrete colors for each wavelength. Please make the legend discrete too.

- Figure 5: What do these boxes and whiskers mean? Please clarify.

- Figure 6: Could you please add the correlation coefficients to the figure?

- Figure S6: I guess you meant $\alpha$ as a function of $\lambda$ or do you mean only the wavelength pair 370-880 nm?

- Figures S4, S7, and S9: Please stick to journal guidelines and avoid the use of the jet (or rainbow) color map: "For maps and charts, please keep colour blindness in mind and avoid the parallel usage of green and red. For a list of colour scales that are illegible to a significant number of readers, please visit ColorBrewer 2.0".

---

## Referee Comment (RC2) · Anonymous Referee #2 · 3 Jul 2018

Review of: Production of particulate brown carbon during atmospheric aging of wood-burning emissions by Kumar et al.

Overall, I find that the authors have presented novel and interesting results on the influence of photochemical aging on absorption by residential wood combustion emissions. They have done a better job than I often see in assessing the performance of the aethelometer for their specific situation, but should include additional details regarding measurement uncertainties, and how these measurement uncertainties propogate to their final atmospheric implications. I have numerous comments, mostly just asking for clarification. I believe this paper should be publishable, with revisions.

Title: It would be good to state "residential wood-burning emissions."

L14: It might be good to indicate this was under (likely) high NOx conditions.

L15: It would be good to clarify what wavelength, or if this is integrated in some way.

L17: Reporting the GSD is informative, but an actual uncertainty estimate would be better.

L54: It would be good to also cite the work from the Georgia Tech group (*Forrister et al.*, 2015).

L229: It is not clear to me why the MAC for POA would be unaffected. The authors write for Eqn. 13 what amounts to:

$$\alpha(\lambda, 880) = \frac{-1}{\ln\left(\frac{\lambda}{880}\right)} \ln\left(\frac{[EC]MAC_{BC_\lambda}}{[EC]MAC_{BC_{880}}} + \frac{[OA]MAC_{OA_\lambda}}{[EC]MAC_{BC_{880}}}\right)$$
$$= \frac{-1}{\ln\left(\frac{\lambda}{880}\right)} \ln\left(\frac{[EC]MAC_{BC_\lambda}}{[EC]MAC_{BC_{880}}} + \frac{[OA]MAC_{OA_\lambda}}{b_{abs,880}}\right)$$

and where they have only included the second line, not the first. The $b_{abs,880}$ value, while seemingly independent of the EC measurements as the authors have written them, actually do depend on the EC measurements because everything has been referenced to the EC measurement. Perhaps I am simply missing something, but I think that the authors statement must be further justified. Yes, the MAC of OA is fundamentally independent of the MAC of BC. But I am not certain that these are practically separated to the extent indicated by the authors. This is the same challenge that all AAE extrapolation methods face in quantitatively determining OA absorption in the presence of BC. See e.g. the cited Moosmuller paper or (*Lack and Langridge*, 2013). Also, the statement on L231 is self-evident. Of course the AAE at a given wavelength depends on the relative contributions of BC and OA.

L264: What about chemically induced changes of POA mass, as opposed to just absorption?

L287: It is unclear what is meant by "a higher than measured lensing effect."

L290: The authors mention uncertainties here for the UV-Vis measurements. But what about for all of the in situ or other measurements? This includes [EC], [OA], absorption. Further discussion of uncertainties is necessary.

L305: It is not clear how this uncertainty estimate was arrived at. Also, this differs from the figure. Finally, it is not clear whether this fit has been forced to zero or not.

L307: Bond et al. did not "report" a value at 880 nm. They reported at a shorter wavelength. This value is inferred assuming an AAE = 1. It should be noted as such.

L308: While the authors obtain a value of the MAC in good agreement with extrapolated values from Bond et al., it is not clear to me how this definitely indicates no lensing effect is present. An uncertainty analysis is necessary. What if, hypothetically, the EC was biased high and the absorption biased low? The obtained MAC might appear in agreement with literature, but only within the bounds of the measurement uncertainty. This statement should be quantified.

L315: It is clear that the distribution is reasonably log-normal with a single mode when considered in number space. But what about in surface area or in volume space, which is important for the calculation of the $MAC\_OA\_bulk$ (Eqn. 20)? Also, to what extent does the volume-weighted distribution exceed the SMPS bounds?

L325 and Fig. S2: I do not understand why in Fig. S2 it says the "reported value" is 2.6 while here 3.0 is given. This should be clarified.

L332: Not only wood combustion, but really any aerosol if it is assumed that the true MAC for BC at 880 nm is ~4.7 m2/g.

L336: This seems circular to me, if the authors are using C = 3 in coming to this conclusion. This is demanded through the various inequalities.

Fig. S5 must indicate which studies are being used to define the literature bounds, and also note that this is not the entire range of reported values over the many papers on this topic. This is a subset of values. Consider e.g. (*Lewis et al.*, 2008) or (*Liu et al.*, 2013). The authors have selected a very narrow subset of literature results to present here, and to reference in the text.

Fig. 3: It is not clear why the data in Fig. A do not overlap with the data in Fig. B. Presumably the data in figure B evolved from the data in Fig. A. Also, in Fig. A is is not clear if each point is for one experiment or whether the variability in f_OA is due to variability within an experiment.

L344: I do not dispute that the AAE values increase with f_OA. However, it is evident that as wavelength decreases the difference from AAE = 0.9-1.1 and the observations increases. This is not clear from the discussion here.

L346: The range reported is inconsistent with what is shown in the graph.

L350: this could be strengthened simply by showing a graph of the observations as a function of wavelength, and showing that a single AAE value does not provide for a good fit.

Fig. 2: For consistency, it would be helpful if the color scale were labeled as f_OA, similar to fig. 3 and the text.

L356: Looking at Fig. 3B, it is not clear that this is generally the case. The highest f_OA in Fig. 3B does not have the highest AAE. Perhaps the uathors mean this just for the high OH exposures. If so, they might consider plotting AAE vs. f_OA for subsets of data binned according to OH exposure. But, as presented, it is not evident that this is a fully general conclusion.

L362: While I don't necessarily disagree with this point, I will reiterate that the relationship between Fig. 3A and 3B is not clear. The authors give a dashed curve, but it is not clear how, for example, the data in Fig. 3B that start at such low AAE values at high f_OA values come from Fig. 3A. A stronger connection needs to be made to make this clearer

L364: For the data in Fig. 3B, extrapolation to f_OA → 0 for wavelengths < 600 nm suggests an AAE ~1.2-1.3, larger than the 0.9-1.1 range the authors have taken for BC. This is consistent with the derived MAC(370) = 13.7 m2/g for BC, given the value at 880 nm. There is, however, a bit of an inconsistency with how the authors compare with Bond. They state that 13.7 is within the 95% confidence interval of the 11.1 m2/g value reported by Bond. But, they have also stated that the AAE = 0.9-1.1. If this is the case, then isn't the range actually narrower? Really, my question here is about the consistency of the statistical interpretation/uncertainty representation.

Fig. 7: I find the legend to be incomplete in that it leaves the reader thinking that the Lu and Saleh measurements are from methanol extraction, which they are not. Also, for Saleh (2014), the authors do not at all make clear that the Saleh measurements are at 550 nm, not 370. This is not a fair comparison. Neither is which fuel type of Saleh's has been considered. The authors should provide a fuller picture. Also, the Saleh reference is missing from the bibliography.

Fig. S7: It is not clear why the propagated uncertainty in the AAE increases with wavelength or f_OA. The AAE is a measured quantity that depends only on the measured absorption at two wavelengths. Why would uncertainty in absorption depend on f_OA? And are the authors saying that the uncertainty in absorption increases with wavelength? Uncertainty in the AAE should directly propagate from Eqn. 10, which is independent of f_OA. Perhaps I am misunderstanding?

L384: A larger GSD does not necessarily mean a larger uncertainty. This could be variability that is independent of uncertainty. I do not regard this as a true assessment of "uncertainty." It is only an assessment of variability. The authors should, however, consider uncertainties explicitly.

L386: Given that the authors show distributions and fits for the AAE, it would seem appropriate to also show similar for the MAC_POA and MAC_SOA so that the reviewer can judge. Given the width of the

SOA distribution, is a normal fit even appropriate? (Probably not, in a fundamental sense, since MAC values cannot be < 0. But perhaps a normal distribution is appropriate in a practical sense.)

L398: The authors should provide the resulting uncertainties, or at least ranges, based on the multivariate analysis, for the AAE values.

L402: I suggest removing the "this is the first study" statement. Saleh (2014) reported very closely related "w" values, from which AAE values can be calculated, for SOA from biomass burning.

Fig. S10: I find that the use of the log scale for the y-axis makes it difficult for the reader to see what sort of changes did/did not occur. Variability in the AAE over so many orders of magnitude is not expected, but a factor of 2 would be reasonable. Thus, a linear scale should be used.

L424: In Fig. S13, and Fig. 6, it is unclear why the authors fit only the "aged" data. Why exclude the primary, especially in Fig. S13? Because the relationship is visibly much worse? This goes to the statement about sensitivity to Mie calculations.

L427: Fig. 6 normalizes out any uncertainty/variability in the measured [OA], because both absorption values are normalized by this. Fig. 4, in contrast, does not. How can the authors rule out the possibility that there is not some vriability in the measurement of OA between burns, perhaps dependent on particle shape or variability in bounce in the AMS (which can differ between POA and SOA)?

L432: This 46% must state that it is for aged OA only. It remains unclear to me why the primary is excluded.

L440: Is this a fair comparison, given that the authors have focused on the aged OA?

L431: Are these fits forced through zero?

L441: The authors seem to be implying that SOA formed from oxidation of aromatic precursors is not especially water soluble, or at least less soluble than in methanol. The authors might consider citing e.g. (*Zhang et al.*, 2011), to strengthen this argument.

Fig. 7: As already noted above, I find the comparison here insufficient. Saleh et al. (2014) and Lu et al. (2015) report values not at 370 nm. This is unclear. Also, the line shown for Lu et al. (2015) appears to be incorrect. See their Fig. 1D. Further, and importantly, the Lu et al. (2015) data are largely, although not entirely, derived from the Saleh measurements. Thus, they are not really an independent assessment.

Fig. 7: The logic of a linear fit to the observations is not clear to me. The authors have argued that the SOA is absorbing, and differently absorbing than the POA. If I use the equation given and extrapolate towards $M\_BC/M\_OA \rightarrow 0$, the $k\_OA \rightarrow 0$. If the SOA is absorbing, and if SOA formation drives the decrease in the $M\_BC/M\_OA$, then the limiting value of $k\_OA$ should be equal to the value for $k\_SOA$. As

such, the provided fit does not seem appropriate and requires justification. Some of this may be experiment-to-experiment variability. But the limiting case issue remains.

Fig. 7: The authors should be able to, from their observations and within their assumptions, calculate M_POA/M_SOA. They might consider plotting k_OA vs. this ratio instead of versus M_BC. These will be related, of course, since the authors assume POA is proportional to BC during aging for a given experiment.

L471: The origin of these "uncertainties" is unclear. They are explained later for f_OA, but for the MAC values it is not abundantly clear.

L486: This statement regarding mass yields of SOA requires much further detail.

L512: A note about terminology. I am not certain that "error analysis" is appropriate here. Variance in the POA fraction is not "error." It is variability. A substantial aspect of this "error analysis" is really just a "sensitivity analysis." I suggest that the authors limit the term "error analysis" to when they are truly considering errors, and use some other term when they are considering variability. This is true here and elsewhere.

L499: The authors should clarify the origin of the solar irradiance data that they have used.

Forrister, H., et al. (2015), Evolution of brown carbon in wildfire plumes, *Geophysical Research Letters*, *42*(11), 4623-4630, doi:10.1002/2015GL063897.
Lack, D. A., and J. M. Langridge (2013), On the attribution of black and brown carbon light absorption using the Angstrom exponent, *Atmospheric Chemistry and Physics*, *13*(20), 10535-10543, doi:10.5194/acp-13-10535-2013.
Lewis, K., W. P. Arnott, H. Moosmüller, and C. E. Wold (2008), Strong spectral variation of biomass smoke light absorption and single scattering albedo observed with a novel dual-wavelength photoacoustic instrument, *Journal of Geophysical Research: Atmospheres*, *113*(D16), D16203, doi:10.1029/2007JD009699.
Liu, S., et al. (2013), Aerosol single scattering albedo dependence on biomass combustion efficiency: Laboratory and field studies, *Geophysical Research Letters*, 2013GL058392, doi:10.1002/2013GL058392.
Zhang, X. L., Y. H. Lin, J. D. Surratt, P. Zotter, A. S. H. Prevot, and R. J. Weber (2011), Light-absorbing soluble organic aerosol in Los Angeles and Atlanta: A contrast in secondary organic aerosol, *Geophysical Research Letters*, *38*, doi:10.1029/2011gl049385.

---

## Author Comment (AC1) · 9 Oct 2018

Dear editor,

We thank both reviewers for their constructive comments, which significantly enhanced the quality of our manuscript. Attached as a pdf is a point-by-point response to the comments of both reviewers.

Please also note the supplement to this comment:
https://www.atmos-chem-phys-discuss.net/acp-2018-159/acp-2018-159-AC1-supplement.pdf

---

## Author Response (AR1)

**Author response to referee's comments for acp-2018-159: Production of particulate brown carbon during atmospheric aging of wood-burning emissions by Kumar et al.**

Dear editor,

We thank both reviewers for their constructive comments, which significantly enhanced the quality of our manuscript. Below, we provide a point-by-point response (regular typeset) to the comments (blue font) of both reviewers. The modifications made to the manuscript are in grey font, indented and *italicized*. Please note that all references to line numbers are to the submitted manuscript (the ACPD file) and not the revised manuscript.

**Anonymous referee 1.**

**GENERAL COMMENT**

The paper presents an analysis of black carbon (BC) and organic aerosol absorption properties upon aging. The experiments were done in a laboratory using a smog chamber where aerosols were exposed to OH radical and UV radiation. Several optical and chemical properties were measured online and filter samples were collected to be analyzed later offline by a multiple wavelength absorbance measurement technique and an EC/OC analyzer. Methanol and water filter extracts were also analyzed and absorbance was measured. The data is very valuable and the paper is well written and presented. I would recommend its publication after addressing the comments I present below.

My main concerns are related to the following aspects:

- The offline techniques used in this study suffer of different artifacts and they are not sufficiently discussed in the manuscript.

We had carefully considered such artifacts, but could have made more reference to them in the manuscript. We consider the reviewer to be referring to unquantifiable uncertainties. Based on both reviewers' comments, we have added a section discussing quantifiable and unquantifiable uncertainties in the method section. This section reads as follows:

> *Uncertainty analysis. It is important to draw a clear distinction between uncertainties related to measurement precision and accuracy and those related with experimental variability. In this section we discuss the quantifiable and unquantifiable uncertainties related with the different measurements. In the result section, we will present our confidence levels on the average parameters determined based on the experimental variability, which we judge to be the main source of variance in the data.*

➔ *Quantifiable uncertainties:*

*The estimated uncertainty in the AMS-derived OA mass concentrations is ~25%, which includes both potential biases and precision. This estimate is based on the variation in the AMS calibration factors and estimated uncertainties in the SMPS used for the AMS calibration (Bruns et al., 2015, 2016). Uncertainties related to particle transmission efficiency in the AMS are considered negligible for the particles sampled here (Liu et al., 2007), whose volume size distribution falls within the range transmitted efficiently by the AMS aerodynamic lens (see Fig. S4). The bounce-related collection efficiency (CE) of the AMS was concluded to be unity for wood-burning OA in the literature reviewed by Corbin et al. (2015b; in their Section S1.2). For the present data, the comparison between the SMPS mass (predicted from fitted volume distributions using a density of 1.5 g cm$^{-3}$) and the total PM predicted as AMS-OA+eBC, suggest a CE value between 0.7 and 1.0 (19% relative uncertainty), consistent with average literature values and the uncertainties estimates. The uncertainty in EC mass concentration, estimated from measurement repeats based on the EUSAAR2 protocol only, is within 7% in our case. The precision uncertainty in the aethalometer attenuation measurements was estimated as 15 Mm$^{-1}$ based on the standard deviation of its signals prior to aerosol being injected into the smog chamber. The MWAA data have an estimated noise level and precision of 12 /Mm and 10% respectively, and these uncertainties have been added in quadrature to provide the overall uncertainties shown, for example, as error bars in Fig. 1 below. To compare the MWAA and aethalometer measurements, we determined $b_{abs,MWAA,880nm}$ by extrapolating the absorption coefficients measured at 850 nm to 880 nm using an α-value determined from the ratio between the absorption coefficients at 850 nm and 635nm. The uncertainty associated with this extrapolation is considered negligible relative to the overall MWAA uncertainty.*

➔ *Possible unquantified uncertainties:*

*There are significant uncertainties in the measurement of aerosol absorption using filter-based techniques (e.g., Collaud Coen et al., 2010). Here, we have used MWAA measurements as a reference to scale the aethalometer data, using a single C value. The correction factor C, which accounts for scattering effects within the filter matrix (Drinovec et al., 2015), may depend on the aerosol sample (Collaud Coen et al., 2010). In this study, we evaluated the variability in this factor for our primary and aged samples, by directly comparing the aethalometer to MWAA measurements, as discussed below. The MWAA has been previously validated against a polar nephelometer and a MAAP (Massabo et al., 2013), which, in turn, has been validated against numerous in situ methods (e.g., Slowik et al., 2007). The excellent correlation between MWAA and EC in our study (discussed below) supports the high confidence in the MWAA filter based absorption measurements conducted here. Another significant source of uncertainty in filter-based absorption measurements is the*

*possible sorption (or evaporation) of volatile organics on (or from) the filter material. This may lead to an overestimation (or underestimation) of OA absorption. However, we have minimized sorption artefacts by utilizing a charcoal denuder. We have obtained an excellent correlation between OA absorption measurements derived from the MWAA-calibrated aethalometer and from quartz filter samples (see discussion below, Fig. 6 in the main text and S13 in the supplementary information). Although both of these techniques involved filter sampling, their sampling timescale is an order of magnitude different, and a difference is therefore expected if sorption (or evaporation) caused a substantial bias in our results. We therefore conclude that it is unlikely that artifacts associated with filter sampling have biased the absorption measurements. Finally, uncertainties related to pyrolysis during thermo-optical analysis may bias EC measurements. Such uncertainties arise from unstable organic compounds, and can be significant for biomass-burning samples, leading to biases on the order of 20% for EC (e.g. Schauer et al., 2003; Yang and Yu., 2007). To minimize these biases we applied the EUSAAR2 protocol. The optical properties of such organics are generally different from BC; therefore, the excellent correlation between MWAA and EC data in Fig. 1A suggest that pyrolysis effects were not a major source of uncertainty in our data set.*

We have added the following abbreviation to the corrected text:

L146. *multi-angle absorption photometer (MAAP, Petzold and Schönlinner, 2004).*

We have also expanded the text in some places to reflect the considerations presented above, as shown in the response to the next comment.

***New references added:***

*Schauer, J. J., Mader, B. T., Deminter, J. T., Heidemann, G., Bae, M. S., Seinfeld, J. H., Flagan, R. C., Cary, R. A., Smith, D., Huebert, B. J., Bertram, T., Howell, S., Kline, J. T., Quinn, P., Bates, T., Turpin, B., Lim, H. J., Yu, J. Z., Yang, H., and Keywood, M. D.: ACE-Asia intercomparison of a thermaloptical method for the determination of particle-phase organic and elemental carbon, Environ. Sci. Technol., 37, 993–1001, https://doi.org/10.1021/es020622f, 2003.*

*Slowik, J. G., E. S. Cross, J.-H. Han, P. Davidovits, T. B. Onasch, J. T. Jayne, L. R. Williams, M. R. Canagaratna, D. R. Worsnop, R. K. Chakrabarty, H. Moosmüller, W. P. Arnott, J. P. Schwarz, R. S. Gao, D. W. Fahey, G. L. Kok and A. Petzold, An inter-comparison of instruments measuring black carbon content of soot particles, Aerosol Sci. Technol. 41, 3, 295-314, 2007.*

*Collaud Coen, M., Weingartner, E., Apituley, A., Ceburnis, D., Fierz-Schmidhauser, R., Flentje, H., Henzing, J. S., Jennings, S. G., Moerman, M., Petzold, A., Schmid, O., and Baltensperger, U.: Minimizing light absorption measurement artifacts of the Aethalometer: evaluation of five correction algorithms, Atmos. Meas. Tech., 3, 457-474, https://doi.org/10.5194/amt-3-457-2010, 2010.*

*Yang, H. and Yu, J. Z.: Uncertainties in charring correction in the analysis of elemental and organic carbon in atmospheric particles by thermal/optical methods, Environ. Sci. Technol., 36 (23), 5199–5204, 2002.*

- In this study the BC particles were not observed to be coated with other kind of particles (i.e., no internal mixing but external mixing was observed). However, ambient studies have shown that BC particles in the atmosphere are usually coated and this coating causes an enhancement of BC absorption. Given that, how representative is this study of "atmospheric aging"? Were the experiments not long enough to "age" the BC particles?

The reviewer raised two points, to which we reply separately.

The reviewer questions how representative our study was of atmospheric aging. The emissions studied here are representative of flaming wood in stoves typically used in Western Europe, while aging is equivalent to ~2 days of OH-driven photochemistry, under atmospheric winter day time conditions in the mid-latitude.

The reviewer questions the reasons for the lack of lensing with aging, when such effect had been observed in the field. We have clarified our language here. Our aim has been to assert that our measurements are poorly represented by a pure core-shell conceptual model of internal mixing. This assertion is based on our measured absorption coefficients, and we have modified the text to explain this in more detail as quoted immediately below.

The AMS measurements showed that the amount of OA generated during aging was substantial. Likewise, the SMPS showed a considerable growth of the primary particles with aging. If BC and OA are naively treated as core-shell mixtures, an absorption enhancement of ~1.8 would have been predicted, with an average increase in the coating mass by a factor of 3 (see Bond et al., 2006). However, our absorption-coefficient measurements in Fig. 1a showed that we did not observe any absorption enhancement. Therefore, we do not conclude that "BC particles were not coated" but rather than "the particles studied could not be represented by a core–shell description of coatings that envelop the central BC core". The particles may be internally mixed, but of a morphology more complex than core-shell − e.g. off-centered coatings with complexities due to the aggregated morphology of BC, see e.g. the microscopy images of biomass-burning particles by China et al. 2013. Modelling or even accurately describing such morphologies is well beyond the scope of our experimental study. Current literature reports for the lensing effect are conflicting, showing that absorption enhancements upon significant BC coating can be less than 5% (Cappa et al., 2012) or as large as 150% (Liu et al., 2015). Recent experimental work suggests that such discrepancies are related to the complex black carbon morphology and a core–shell description does not adequately capture mixed-BC optical properties and may considerably overestimate the observed absorption values (Liu et al., 2017). As lensing effect was negligible in our case, we have assumed that the aerosol optically behaves as an external mixture between BC and BrC. We note that while this assumption is important for estimating the BC absorption, the BrC absorption is not very sensitive to the assumed morphology. Based on both reviewers comments and to avoid
confusion we have modified the Results section 4.1 as follows:

*Section 4.1.*

[revised manuscript text omitted]

We slightly modified Fig. 1A to reflect these changes. We included error bars in Fig. 1 which
had been missing previously.

[Figure]

We also included error bars in Fig. 7:

[Figure]

SPECIFIC COMMENTS

1.      Lines 107-108: How can you guarantee the correction factor C is wavelength independent?

This is a good question which has often been neglected in the literature. Recently, Corbin et al. (2018), in their Section S3.2, presented a very detailed discussion and reanalysis of the wavelength dependence of the C-value.

In that publication, the authors described the wavelength dependence of the C-value as separated into a filter dependence and a scattering cross-sensitivity measurement, and presented four different arguments for its wavelength dependence being negligible:

1.  They compared their wavelength-dependent absorption coefficient measurements with MWAA measurements (which do not rely on a C-value-like correction) and found good agreement between the two techniques. We have also verified our AE33 data with MWAA data in the present study.
2.  They pointed out that the measured AAE would be biased by a wavelength-dependent C value, so that their measurements of an AAE of 1.0 for samples dominated by BC (in agreement with extensive literature) indicates a negligible wavelength dependence of C.
3.  They combined measurements of aerosol SSA (ranging from 0.5 to 0.9) with size-dependent scattering cross-sensitivity measurements to quantitatively estimate the influence of scattering cross-sensitivity as negligible. SSA measurements were not available in our study, but our measured size distributions indicate that our particles were generally small enough that their conclusions can be extrapolated to our samples.
4.  They described previous work where different filter materials were compared, with no significant effect on the wavelength dependence of the C value (Drinovec et al., 2015).

In conclusion, the C value is known to depend on the filter material but its wavelength dependence has been shown to be negligible for samples such as those studied in the manuscript presently under review.

We have updated the manuscript as follow:

    L111. *As discussed in detail by Corbin et al. (2018), the wavelength-dependence of C can be expected to be negligible*

Based on the referee comment, we have changed the three figures as follows:

[Figure]

**Fig. S4**

[Figure]

**Fig. S7**

[Figure]

**Fig. S9**

**Anonymous referee 2.**

**GENERAL COMMENT**

Overall, I find that the authors have presented novel and interesting results on the influence of photochemical aging on absorption by residential wood combustion emissions. They have done a better job than I often see in assessing the performance of the aethelometer for their specific situation, but should include additional details regarding measurement uncertainties, and how these measurement uncertainties propogate to their final atmospheric implications. I have numerous comments, mostly just asking for clarification. I believe this paper should be publishable, with revisions.

We thank the referee for her/ his constructive comments, which we address below.

1.      Title: It would be good to state "residential wood-burning emissions."

The title has been modified in the corrected version of the manuscript:

> *Production of particulate brown carbon during atmospheric aging of residential wood-burning emissions*

2.      L14: It might be good to indicate this was under (likely) high NOx conditions.

The experiments were conducted at estimated $NO_x$/NMOG ratios of ~ 0.035 – 0.35 ppm ppm $C^{-1}$ (Bruns et al., 2016). These conditions can be considered as high $NO_X$, where most of the $RO_2$ radicals react with NO, rather with $RO_2$/$HO_2$. This information has been added to the corrected version of the manuscript in section 2.1.

> *Section 2.1.*
> *Laboratory measurements were conducted in an 8 $m^3$ Teflon smog chamber (Bruns et al., 2015; Platt et al., 2013) installed within a temperature-controlled housing. Conditions in the chamber were maintained to represent winter time in Europe, i.e. relative humidity ranging between 50 – 90%, at 263 K (Bruns et al., 2015, 2016). Beech wood was combusted in a residential wood stove. Primary emissions were sampled through heated lines at 413 K, diluted by a factor of ~14 using an ejector diluter (DI-1000, Dekati Ltd.), then sampled into the chamber, which provided an additional ten-fold dilution. The overall dilution was a factor of 100 to 200. As we aimed to sample only flaming-phase emissions into the chamber, samples were taken when the modified combustion efficiency (ratio of $CO_2$ to the sum of CO and $CO_2$) was > 0.90. Despite maintaining the same combustion conditions, the resulting organic fraction to the total carbonaceous aerosols in the different samples was*

*highly variable, indicating that these samples are representative of a mixture of pre-ignition and flaming emissions (with varying contributions of each combustion stage). Finally, the resulting NOx/NMOG ratios, which dramatically influence SOA formation through influencing the fate of peroxy radicals, $RO_2$, were estimated to be between 0.035 – 0.35 ppm ppm $C^{-1}$ (Bruns et al., 2016). These conditions can be considered as high $NO_X$ representative of urban/sub-urban conditions, where most of the $RO_2$ radicals react with NO, rather with $RO_2/HO_2$.*

3. L15: It would be good to clarify what wavelength, or if this is integrated in some way.

This has been clarified in the corrected version of the manuscript:

*At shorter wavelengths (370 – 470nm), light absorption by brown carbon from primary organic aerosol (POA) and secondary organic aerosol (SOA) formed during aging was around 10 % and 20 %, respectively, of the total aerosol absorption (BrC plus BC).*

4. L17: Reporting the GSD is informative, but an actual uncertainty estimate would be better.

Based on this comment and others below, the reviewer is asking for providing uncertainty propagation based on measurement precision and accuracy, instead of a GSD which represents the experimental variability. As we discussed in the reply to the first reviewer, we have added in the corrected version of the manuscript a new section in the Method, discussing quantifiable and unquantifiable uncertainties. However, in the result section we still present the variability in the parameters determined as GSD, as these represent our confidence levels in these average parameters. We believe that such information is relevant if these parameters were to be used for future predictions. We also consider that for most of the parameters experimental variability is much more important than measurement uncertainties and biases.

5. L54: It would be good to also cite the work from the Georgia Tech group (*Forrister et al.*, 2015).

As suggested by the reviewer the work of Forrister et al., 2015 was cited.

*Forrister, H., Liu, J., Scheuer, E., Dibb, J., Ziemba, L., Thornhill, L. K., Anderson, B., Diskin, G., Perring, A. E., Schwarz, J. P., Campuzan-Jost, P., Day, D. A., Palm, B. B., Jimenez, J. L., Nenes, A., Weber, R. J.: Evolution of brown carbon in wildfire plumes, Gephys. Res. Lett.,42, 4623-4630, doi: 10.1002/2015GL063897, 2015.*

6.      L229: It is not clear to me why the MAC for POA would be unaffected. The authors write for Eqn. 13 what amounts to:

$$\alpha(\lambda, 880) = \frac{-1}{\ln\left(\frac{\lambda}{880}\right)} \ln\left(\frac{[EC]MAC_{BC_\lambda}}{[EC]MAC_{BC_{880}}} + \frac{[OA]MAC_{OA_\lambda}}{[EC]MAC_{BC_{880}}}\right)$$

$$= \frac{-1}{\ln\left(\frac{\lambda}{880}\right)} \ln\left(\frac{[EC]MAC_{BC_\lambda}}{[EC]MAC_{BC_{880}}} + \frac{[OA]MAC_{OA_\lambda}}{b_{abs,880}}\right)$$

and where they have only included the second line, not the first. The $b_{abs,880}$ value, while seemingly independent of the EC measurements as the authors have written them, actually do depend on the EC measurements because everything has been referenced to the EC measurement. Perhaps I am simply missing something, but I think that the authors statement must be further justified. Yes, the MAC of OA is fundamentally independent of the MAC of BC. But I am not certain that these are practically separated to the extent indicated by the authors. This is the same challenge that all AAE extrapolation methods face in quantitatively determining OA absorption in the presence of BC. See e.g. the cited Moosmuller paper or (*Lack and Langridge*, 2013). Also, the statement on L231 is self-evident. Of course the AAE at a given wavelength depends on the relative contributions of BC and OA.

The reviewer raises two separate points. The first is whether the $MAC_{OA,\lambda}$ directly depends on the EC mass. The second is whether the BC absorption at a given $\lambda$ affects the estimation of $MAC_{OA,\lambda}$. Below, we address these points separately.

1. We consider it inaccurate to say that OA MAC has been referenced to EC. This reflects how we have expressed and applied Equation 13. The intermediate steps leading to equation 13 were intentionally omitted, as they tend to mislead the reader. Equation 13 clearly shows that:
   - EC mass concentration is not explicitly required.
   - A potential bias in $MAC_{BC,880nm}$ due to a bias in EC mass would directly affect the resulting $MAC_{BC,\lambda}$ in a proportional manner, whereas $MAC_{OA,\lambda}$ would remain completely unaffected.
   - The resulting $MAC_{OA,\lambda}$ depends on the input parameter $M_{OA}$, thus being affected by potential AMS calibration bias.
   - A potential bias in absorption coefficients measured by the MWAA would propagate to a proportional bias in $MAC_{OA,\lambda}$, as aethalometer measurements of $b_{abs}$ are referenced against the MWAA. Such bias in absorption coefficient would also propagate to a proportional bias in $MAC_{BC,\lambda}$, which would happen through a corresponding bias in $MAC_{BC,880}$
   - The resulting $MAC_{OA,\lambda}$ also depends on the input parameter $b_{abs,880nm}$, which is referenced to the MWAA measurement, whereas the EC data are by no means blended into the $b_{abs}$ data. Such bias in absorption coefficient would also propagate to a proportional bias in $MAC_{BC,\lambda}$, which would happen through a corresponding bias in $MAC_{BC,880nm}$.

2. We acknowledge that the determination of $MAC_{OA,\lambda}$ depends on the estimated absorption of BC at a given $\lambda$, which in turn depends on the estimated $MAC_{BC,\lambda}$. This is an issue on any multivariate fitting, where the theoretically independent fitted quantities are not independently determined. We note that such uncertainties are taken into account by the fitting errors presented as GSDs in the manuscript. The $MAC_{BC,\lambda}$ is similar to an extrapolation of the absorption measurements at $f_{OA} = 0$. We note our experiment covered BC rich particles allowing for an accurate determination of $MAC_{BC,\lambda}$.

The reviewer raises a valid point that unidentified measurement biases in opposing directions may have led to the illusion of agreement between our measured $MAC_{BC}$(880nm) values and the literature values for bare BC. The major issue here is that of referencing our absorption data and EC measurements to reliable and calibrated technique. Absorption measurements were obtained using MWAA, which has been validated as described in the text, and biases are expected to be within 10%. Measurement biases related to total carbon measurement are negigible (within 5%). The high correlation between absorption and EC measurements also indicates that unquatifiable uncertainties and artefacts (e.g. charring for EC and vapor adsorption artefacts for MWAA) are negligible, as the fundamental differences between the two measurements mean that any artefacts are unlikely to be similar between them. Therefore, measurement biases and unquatifiable artefacts are unlikely to affect the estimated values for $MAC_{BC}$(880nm) and our conclusions about the lack of lensing. We also note that such conclusions are also supported by independent time resolved attenuation measurements by the aethalometer, suggesting that little (<10%) to no increase in the attenuation coefficients upon SOA formation.

Our analysis has combined multiple analytical techniques and found good agreement between all of them. This good agreement reduces the likelihood that opposing measurement biases led, by chance, to our measurements being in agreement with literature. While it remains theoretically possible that unknown uncertainties and biases existed in our analysis, it is by Occam's razor more probable that our measurements were in fact accurate and that our cross-validation by employing different techniques was successful.

In the corrected version of the manuscript, we have added a new section discussing the measurement biases estimated for the different measurements. We have additionally discussed potential unquantifiable uncertainties. For clarity, we do not duplicate the mofified text here, but we quote the response to reviewer 1.

13.    L315: It is clear that the distribution is reasonably log-normal with a single mode when considered in number space. But what about in surface area or in volume space, which is important for the calculation of the MAC_OA_bulk (Eqn. 20)? Also, to what extent does the volume-weighted distribution exceed the SMPS bounds?

The absolute SMPS volume is not as important for calculating $MAC_{OA,bulk}$ as the reviewer understood. As can be seen from Equation 20, we did not use the SMPS data to calculate total OA volume. We measured total OA mass with an AMS, converted mass into volume using an assumed density (the assumed density has no impact on the results as we use the same density to calculate MAC), and then "distributed" this volume across the size distribution measured by SMPS. That is, the SMPS data provide only a weighting factor for the size dependence of absorption, which means that uncertainties in these data do not have a major effect on our results. We performed the calculations in this way to minimize associated uncertainties, but acknowledge that substantial uncertainties may result. We have estimated that these may be on the order of 20%, based on the magnitude with which particulate absorption varies as a function of size (according to Mie theory).

14.    L325 and Fig. S2: I do not understand why in Fig. S2 it says the "reported value" is 2.6 while here 3.0 is given. This should be clarified.

The reported C value is calculated using a $\sigma_{ATN} = 12.2$ m$^2$g$^{-1}$, as given by the manufacturer, and $MAC_{BC}(880nm) = 4.7$ m$^2$ g$^{-1}$. The C value = 3 is determined from our attenuation and absorption measurements which is used in our calculations. This has been discussed in lines 330-336.

For better clarity, we have now omitted the reported C-value from Fig. S2:

[Figure]

*Figure S2: (A) Probability density function (PDF) comparing the MAC values determined by normalizing MWAA absorption measurements of offline primary (filter A), slightly aged (filter B: Aged1) and aged (filter C: Aged2) samples to EC (EUSAAR2) measurements of the same samples (bold line). A literature value for pure BC is also shown (Bond et al., 2006) (dashed blue line). (B) PDF comparing aethalometer attenutation measurements at 880 nm and MWAA absorption measurements at 850 nm to retrieve the aethalometer C value.*

15.    L332: Not only wood combustion, but really any aerosol if it is assumed that the true MAC for BC at 880 nm is ~4.7 m2/g.

We agree with the reviewer. We specify wood burning because it is the focus of our study. This section has been substantially modified and this sentence was removed.

16.    L336: This seems circular to me, if the authors are using C = 3 in coming to this conclusion. This is demanded through the various inequalities.

We do not think that the math here is circular. Below, we present in bullet points the approach followed:

- $\sigma_{ATN} = MAC_{BC} * C$
- The Aethalometer provides $b_{ATN}$
- We determine a C value of 3 from Equation 1 of the paper:

$$C = b_{ATN,AE33} / b_{abs,MWAA}$$

- We determine a $MAC_{BC}$ of 4.6 m$^2$ g$^{-1}$ by a fit through $b_{abs,MWAA}$ and EC thermo-optical measurements for primary and aged filter samples.
- We have therefore determined the two variables required to calculate $\sigma_{ATN}$ and determined it as 13.8 m$^2$ g$^{-1}$.
- We then compare the $\sigma_{ATN}$ determined by us to the manufacturer value of 12.2 m$^2$g$^{-1}$.
- We then concluded that while the factory default $b_{abs}(\lambda)$ has a substantial bias the eBC mass determined using default $\sigma_{ATN}$ is consistent with the EC mass. We have rewritten Section 4.1, as quoted at the beginning of this response, to clarify this reasoning.

 Fig. S5 must indicate which studies are being used to define the literature bounds, and also note that this is not the entire range of reported values over the many papers on this topic. This is a subset of values. Consider e.g. (*Lewis et al.*, 2008) or (*Liu et al.*, 2013). The authors have selected a very narrow subset of literature results to present here, and to reference in the text.

While we agree with the reviewer here, one reason that we have not cited e.g. Lewis et al (2008) is that those authors did not experiment on log-wood burning in a modern stove but rather simulated wildfires. Considering the very low emphasis placed on Fig. S5 in our manuscript, and the importance of avoiding confusion between log-wood burning and wildfire burns, we have simply removed the α ranges in Fig. S5 is only for primary biomass burning emissions. We have also modified the legend to show the two wavelengths of α.

18. Fig. 3: It is not clear why the data in Fig. A do not overlap with the data in Fig. B. Presumably the data in figure B evolved from the data in Fig. A. Also, in Fig. A it is not clear if each point is for one experiment or whether the variability in f_OA is due to variability within an experiment.

Fig. 3A refers to primary emissions and Fig. 3B refers to secondary emissions (as indicated in the caption), so yes, the presumption here was correct, but only partially. More data are included in Fig. 3A because not all burn experiments were aged. This information has been added to the figure caption. In Fig. 3A each point represents an experiment; therefore the variability in $f_{OA}$ is due to the variability in emission composition between experiments. Meanwhile, in Fig. 3B the variability in $f_{OA}$ is due to SOA formation with aging; data from several experiments are shown which explains the wide range of $f_{OA}$ at low OH exposures. Based on the reviewer comment, we have added the following clarifications to the Fig. 3 caption:

> *Figure 3: (A) Relationship of $\alpha_{BC+POA}$ (λ, 880nm) to $f_{OA}$ for seven wavelengths for primary emissions. Data are colored by the wavelength. Curves are fits of Equation 13 to the data. Each point represents the average of one experiment and therefore the variability in $f_{OA}$ is related to the variability in the emission composition between experiments. (B) Relationship of $\alpha_{BC+POA+SOA}$(370nm, 880nm) to $f_{OA}$ for several experiments for aged aerosols. Data are color coded by the OH exposure. The variability in $f_{OA}$ is due to SOA formation with aging; data from several experiments are shown which explains the wide range of $f_{OA}$ at low OH exposures. Note that more data are included in A than B, as primary emissions for some experiments were not aged.*

19.     L344: I do not dispute that the AAE values increase with f_OA. However, it is evident that as wavelength decreases the difference from AAE = 0.9-1.1 and the observations increases. This is not clear from the discussion here.

We have stated that the AAE values at low $f_{POA}$ are close to those reported for pure BC. We note that for $\lambda = 660$ nm, the AAE value extrapolated at $f_{POA} = 0$, is equal to 1.04, while for all other wavelengths the value is statistically similar, equal to ~1.2. It can be seen from Fig. 4 that $MAC_{BC}(\lambda)$ profile can be clearly described by a power law, consistent with a constant AAE. To avoid confusion, we have modified the text as follows:

> L344. *The $\alpha(\lambda, 880nm)$ is slightly higher than that of pure BC (~1.2; Bond et al., 2013; Zotter et al., 2017) for small $f_{POA}$, while increasing $f_{POA}$ corresponded to a distinct increase of $\alpha(\lambda, 880nm)$.*

20.     L346: The range reported is inconsistent with what is shown in the graph.

The range presented before denoted the P10 and P90; we apologize that we have forgotten to mention this in the text. Based on the reviewer comment and to avoid confusions we have replaced the [P10-P90] by the total range:

> *The $f_{POA}$ ranges from **0.12 to 0.63**, which is lower than $f_{POA}$ reported for open burning emissions (e.g., $f_{POA}$~0.75, Ulevicius et al (2016)), because our wood-stove emissions feature a more efficient combustion.*

21.     L350: this could be strengthened simply by showing a graph of the observations as a function of wavelength, and showing that a single AAE value does not provide for a good fit.

This is shown in Fig. S5 of the SI. We have added in the corrected version of the manuscript a reference to Fig. S5 and modified the figure caption to highlight the point raised by the reviewer as follows:

> *Figure S5: Absorption coefficients of fresh wood burning emissions measured using an aethalometer normalized to the eBC mass as a function of wavelength. In the legend each color denotes the $\alpha_{BC+POA}(370nm,880nm)$ for an individual experiment. The dashed lines mark the absorption profiles calculated using the literature range of $\alpha$ values obtained for primary biomass burning emissions. The observed absorption spectra have steeper gradients with decreasing wavelength compared to the lines of constant alpha. The systematic decrease in $\alpha(\lambda, 880nm)$ with increasing $\lambda$ reflects the more-efficient light absorption by BrC at shorter wavelengths (Moosmüller et al., 2011), and shows that the power law wavelength dependence is an inaccurate oversimplification for this mixed aerosol.*

We have added the following in the text in L350:

*As illustrated in Fig. S5, the observed absorption spectra have steeper gradients with*
*decreasing wavelength compared to the lines of constant alpha. Such systematic*
*increase in $\alpha(\lambda, 880nm)$ with decreasing $\lambda$ reflects the more-efficient light absorption*
*by BrC at shorter wavelengths (Moosmüller et al., 2011), and shows that the power*
*law wavelength dependence is an inaccurate oversimplification for this mixed*
*aerosol.*

22.     Fig. 2: For consistency, it would be helpful if the color scale were labeled as f_OA,
similar to Fig. 3 and the text.

This has been modified in the corrected version of the manuscript:

[Figure]

23. L356: Looking at Fig. 3B, it is not clear that this is generally the case. The highest f_OA in Fig. 3B does not have the highest AAE. Perhaps the authors mean this just for the high OH exposures. If so, they might consider plotting AAE vs. f_OA for subsets of data binned according to OH exposure. But, as presented, it is not evident that this is a fully general conclusion.

We do mean at higher OH exposures. This has been added in the corrected version of the manuscript:

> *Also, note in Fig. 2 that at highest OH exposures, the highest $\alpha_{BC+POA+SOA}$(370nm, 880nm) were reached, on average 1.8, during experiments where the fOA was highest.*

24. L362: While I don't necessarily disagree with this point, I will reiterate that the relationship between Fig. 3A and 3B is not clear. The authors give a dashed curve, but it is not clear how, for example, the data in Fig. 3B that start at such low AAE values at high f_OA values come from Fig. 3A. A stronger connection needs to be made to make this clearer

We think that the misunderstanding comes from the fact that we had not adequately highlighted that not all experiments in Fig. A are shown in Fig. B, as for some of the experiments the emissions were not aged. We chose to represent use all the data available in Fig. A to increase our statistics and expand the $f_{OA}$ range. As mentioned above this information has been added and the Figure caption now reads as follows:

> *Figure 3: (A) Relationship of $\alpha_{BC+POA}$ (λ, 880nm) to $f_{OA}$ for seven wavelengths for primary emissions. Data are colored by the wavelength. Curves are fits of Equation 13 to the data. Each point represents the average of one experiment and therefore the variability in $f_{OA}$ is related to the variability in the emission composition between experiments. (B) Relationship of $\alpha_{BC+POA+SOA}$(370nm, 880nm) to $f_{OA}$ for several experiments for aged aerosols. Data are color coded by the OH exposure. The variability in $f_{OA}$ is due to SOA formation with aging; data from several experiments are shown which explains the wide range of $f_{OA}$ at low OH exposures. Note that more data are included in A than B, as primary emissions for some experiments were not aged.*

25.    L364: For the data in Fig. 3B, extrapolation to f_OA -> 0 for wavelengths < 600 nm suggests an AAE ~1.2- 1.3, larger than the 0.9-1.1 range the authors have taken for BC. This is consistent with the derived MAC(370) = 13.7 m2/g for BC, given the value at 880 nm. There is, however, a bit of an inconsistency with how the authors compare with Bond. They state that 13.7 is within the 95% confidence interval of the 11.1 m2/g value reported by Bond. But, they have also stated that the AAE = 0.9-1.1. If this is the case, then isn't the range actually narrower? Really, my question here is about the consistency of the statistical interpretation/uncertainty representation.

We thank the reviewer once again for raising opportunities for clarificating the text, where some of the information were missing. The range of $MAC_{BC}(370)$ we calculate is based on an error propagation calculation considering not only the range of AAE reported (0.9-1.1) but also the errors on the absolute $MAC_{BC}(520)$. This has now been clarified in the text:

> *The obtained fit value of $MAC_{BC}(370nm)$ was 13.7 $m^2$ $g^{-1}$ (GSD 1.1, one-sigma uncertainty 12.4—15.1 $m^2/g$), higher but not statistically significantly different from the range estimated based on Bond et al. (2013), considering the uncertainties on both the $\alpha_{BC}$ values and the $MAC_{BC}(520nm)$.*

26.    Fig. 7: I find the legend to be incomplete in that it leaves the reader thinking that the Lu and Saleh measurements are from methanol extraction, which they are not. Also, for Saleh (2014), the authors do not at all make clear that the Saleh measurements are at 550 nm, not 370. This is not a fair comparison. Neither is which fuel type of Saleh's has been considered. The authors should provide a fuller picture. Also, the Saleh reference is missing from the bibliography.

Thank you for pointing out the missing reference. The Saleh et al. (2014) data in Fig. 7 were extrapolated to 370 nm using the wavelength dependence of $k_{OA}$ i.e. $k_{OA} = k_{OA,550}$ x $(550 / \lambda)^w$, given by those authors. Likewise, the Lu et al., 2015 data have also been determined at 370 nm using the wavelength dependence of $k_{OA}$ provided by the authors ($k_{OA} = 0.017$ x $(550 / \lambda)^{1.62}$) and the parameterization of $k_{OA}$ against BC/OA ratio. We have updated the figure legend and caption according to the reviewer comment:

[Figure]

*Figure 7: Imaginary part of the OA refractive index at 370 nm, obtained from offline UV/vis spectroscopy of methanol OA extracts, plotted as a function of $f_{OA}$. The data could be empirically represented by a linear function in the log-log space, in the measurement range. The ordinary least-squares fit is $(k_{OA,nm}) = \log(M_{BC}/M_{OA})(0.51 \pm 0.07) + (-0.98 \pm 0.05)$. Also shown are parameterizations of $k_{OA}(370nm)$ for open burning against $M_{BC}/M_{OA}$ estimated based on the online $k_{OA}(550nm)$ measurements in Saleh et al. (2014) and Lu et al. (2015), using the $k_{OA}$ wavelength dependence reported by the respective authors.*

***Reference added:***

Saleh, R., Robinson, E. S., Tkacik, D. S., Ahern, A. T., Liu,S., Aiken, A. C., Sullivan, R. C., Presto, A. A., Dubey, M. K., Yokelson, R. J., Donahue, N. M., and Robinson, A.L.:Brownness of organics in aerosols from biomass burning linked to their black carbon content, Nat. Geosci., 7, 2–5, doi:10.1038/ngeo2220, 2014.

27.  Fig. S7: It is not clear why the propagated uncertainty in the AAE increases with wavelength or f_OA. The AAE is a measured quantity that depends only on the measured absorption at two wavelengths. . Why would uncertainty in absorption depend on f_OA? And are the authors saying that the uncertainty in absorption increases with wavelength? Uncertainty in the AAE should directly propagate from Eqn. 10, which is independent of f_OA. Perhaps I am misunderstanding?

We believe that there is a misunderstanding. Fig. S7A is obtained from the error propagation of equation 13 solved for different wavelengths, using the geometric mean and standard deviation of $MAC_{POA}(\lambda)$ and $MAC_{BC}(\lambda)$. The resulting error term represents the variability in/ the confidence level on the $\alpha(t_0, \lambda, 880nm)$ at different wavelengths. Equation 13 is expressed below:

$$\alpha(t_0, \lambda, 880nm) = \alpha_{\mathrm{BC+POA}}(t_0, \lambda, 880nm)$$

$$= \frac{1}{\ln(880\mathrm{nm}/\lambda)} \ln\left( \frac{\mathrm{MAC_{BC}}(t_0,\ \lambda)}{\mathrm{MAC_{BC}}(t_0, 880nm)} + \frac{M_{\mathrm{OA}}(t_0)\mathrm{MAC_{POA}}(t_0,\lambda)}{b_{\mathrm{abs}}(t_0, 880nm)} \right)$$

As $\alpha(t_0, \lambda, 880nm)$ depends on $\mathrm{M_{OA}}/b_{\mathrm{abs}}(t_0, 880nm)$ $\sigma_{\alpha(t_0,\lambda,880nm)}$ also does. We expressed $\mathrm{M_{OA}}/b_{\mathrm{abs}}(880nm)$ as $f_{OA}$, using $\sigma_{ATN}$ to estimate EC mass from $b_{\mathrm{ATN}}(880nm)$.

The image plot in panel B shows that at short wavelengths and low fractions of OA, the confidence level on α is within 0.1. However, with increasing $f_{OA}$, and at longer wavelength the uncertainty in predicting α increases. The idea behind this figure is to provide an error on the predicted α when the $f_{OA}$ is extrapolated to values higher than measured here (typical of open burning).

We have updated the figure caption adding the explanations above:

*Figure S7: Analysis of the fitting errors of $\alpha(\lambda, 880nm)$ of primary emissions as a function of $f_{OA}$. Panel A shows the α residual as a probability density function. Panel B is an image plot of the $\alpha(\lambda, 880nm)$ error, $\sigma_{\alpha(t_0,\lambda,880nm)}$, as a function of $f_{OA}$ at different wavelengths. $\sigma_{\alpha(t_0,\lambda,880nm)}$ is obtained from the error propagation of Eq. (13) solved for different wavelengths, using the geometric mean and standard deviation of $MAC_{POA}(\lambda)$ and $MAC_{BC}(\lambda)$. This error term represents the variability in or the confidence level on the $\alpha(t_0, \lambda, 880nm)$ at different wavelengths. As $\alpha(t_0, \lambda, 880nm)$ depends on $M_{OA}/b_{abs}(t_0, 880nm)$ in Equation 13, $\sigma_{\alpha(t_0,\lambda,880nm)}$ also does. We expressed $M_{OA}/b_{abs}(t_0, 880nm)$ as $f_{OA}$, using $\sigma_{ATN}$ to estimate EC mass from $b_{ATN}(880nm)$. At short wavelengths and low OA fractions, the confidence level on α is within 0.1. However, with increasing $f_{OA}$, and at longer wavelength the uncertainty in predicting α increases.*

28. L384: A larger GSD does not necessarily mean a larger uncertainty. This could be variability that is independent of uncertainty. I do not regard this as a true assessment of "uncertainty." It is only an assessment of variability. The authors should, however, consider uncertainties explicitly.

The reviewer is correct, the word "variability" rather than "uncertainty" should have been used in this sentence. That is, the GSD values relate to variability in the MAC values that is not explained by the variability in $f_{OA}$. Overall, our data show that this variability is related to a real change in the chemical nature of the compounds present and their intrinsic absorptivity, as online MAC values correlate well with $k_{OA}$ values independently measured offline after methanol extraction. As mentioned above, we have now added a new section discussing the quantifiable and unquantifiable uncertainties. We have also updated the related to the variability in the determined parameters as follows:

*__Uncertainties and variability in MAC__$_{BC}$__, MAC__$_{POA}$ __and MAC__$_{SOA}$__. Table 1 shows the fitting errors related with MAC__$_{BC}$__($\lambda$), MAC__$_{POA}$__($\lambda$) and MAC__$_{SOA}$__($\lambda$), arising from our measurement precision and experimental variability. These fitting errors are greater than our estimated uncertainties in the absorption coefficients measured by MWAA (10%), and comparable to our estimated uncertainty in OA mass measured by AMS (30%). The residuals in the fitted MAC__$_{BC}$__($\lambda$) are relatively low (< 10%), increasing with decreasing $\lambda$. By contrast, the uncertainties in the fitted MAC__$_{POA}$__($\lambda$) are much higher (GSD = 1.2–1.5) and increase with increasing $\lambda$. The relative residuals between the measured and fitted $\alpha$($\lambda$,880nm) for primary emissions showed a mean bias and RMSE of 0.07 and 0.13, respectively (Fig. S7), indicating that our fitted MAC results provide a good description of the data set. MAC__$_{SOA}$__($\lambda$) values determined were highly variable between experiments with a GSD = 1.39 and 2.42 for $\lambda$=370 nm and 660 nm, respectively. In Fig. S10, we show the distribution of MAC__$_{SOA}$__($\lambda$) values as box and whiskers against OH exposure, showing no particular dependence of these values with aging as it will be discussed below. Therefore, we expect the fitting errors in MAC__$_{SOA}$ __and of MAC__$_{POA}$ __to be mainly related to true changes in the organic aerosol chemical composition between different burns, since the variability of MAC__$_{BC}$__($\lambda$) was relatively small. In Section 4.3, we discuss this variability further using the results of an additional and independent analysis.*

29.	L386: Given that the authors show distributions and fits for the AAE, it would seem appropriate to also show similar for the MAC_POA and MAC_SOA so that the reviewer can judge. Given the width of the SOA distribution, is a normal fit even appropriate? (Probably not, in a fundamental sense, since MAC values cannot be < 0. But perhaps a normal distribution is appropriate in a practical sense.)

The MACs of primary species (POA and BC) at different wavelengths are obtained through fitting equation 13, which used the AAE as dependent variable. Therefore, we have assessed the model goodness of fit by showing the residuals in the AAE values (Fig. S7). While we cannot show the residuals distributions for $MAC_{POA}$ and $MAC_{BC}$ resulting from the fit, the obtained fitting errors (GSD) can be used to represent these distributions. The normal fit of the AAE residuals in Fig. S7A serves only to illustrate the distribution of fitting errors and is not essential to our analysis. We note that we have never claimed that MAC values to be normally distributed and hence we performed our fits in log-transformed space to constrain the MAC to be greater than zero. $MAC_{SOA}$ is the only unknown parameter in equation 19. Therefore, we have shown the distributions of $MAC_{SOA}$ as a function of wavelength and OH exposure, in Fig. 4 and 5, respectively.

30.	L398: The authors should provide the resulting uncertainties, or at least ranges, based on the multivariate analysis, for the AAE values.

The ranges for AAE values have been already provided in Fig. S9 and Table S1, and following the reviewer's comment we have added the following to the main text:

> *"This yielded αBC = 1.2, [...], with corresponding uncertainties of approximately 20% (complete details of the uncertainties are provided in Table S1)."*

31. L402: I suggest removing the "this is the first study" statement. Saleh (2014) reported very closely related "w" values, from which AAE values can be calculated, for SOA from biomass burning.

The new sentence reads as follows:

> *The high α values obtained for the organic fractions are consistent with previous measurements for BrC containing POA (e.g. Chakrabarty et al., 2010, 2013).*

32.    Fig. S10: I find that the use of the log scale for the y-axis makes it difficult for the reader to see what sort of changes did/did not occur. Variability in the AAE over so many orders of magnitude is not expected, but a factor of 2 would be reasonable. Thus, a linear scale should be used.

In response to the reviewer comment, we have modified the figure to better illustrate the variability in our data.

[Figure]

33. L424: In Fig. S13, and Fig. 6, it is unclear why the authors fit only the "aged" data. Why exclude the primary, especially in Fig. S13? Because the relationship is visibly much worse? This goes to the statement about sensitivity to Mie calculations.

We apologize; this information went missing during revisions of an earlier manuscript draft. We excluded the primary because our assumption of particle sphericity inherent in the Mie calculations is generally more likely to be violated for fresh than for aged combustion particles. Similarly, our interpretation of the SMPS-measured mobility diameter as representative of a physical particle diameter is violated in the presence of fractal-like soot particles, which have a shape factor significantly different from unity.

The variability in our primary results is interpreted as illustrating directly the impact of this variability on our analysis. This variability will reflect differences in burn conditions as well as the chaotic impacts of the combustion process (for example, uncontrolled differences between time spent in pre-ignition [where most OA is emitted] versus flaming phases [where most BC is emitted], between the physical distribution of flames during each burn, etc.).

In spite of these simplifications, the fresh data in Fig. 6 generally fall on the best-fit line for the aged data. The fact that they fall on the best-fit line is a direct demonstration of the magnitude of the uncertainties in our retrieved OA MAC.

We have revised the figures and captions for clarity. The revised figures are:

[Figure]

We modified the text:

L431. *The data in Fig. 6B show that the methanol extracts correspond to a MAC*
*about 50% smaller than the online data. The scatter in the data is significantly*
*reduced for the aged data (note that, in this analysis, aged OA refers to the sum of*
*POA and SOA, since the reported values represent all OA after aging). This reduced*

*scatter is expected, considering that aging is likely to result in more-spherical particles. We have assumed particle sphericity when interpreting the SMPS data and performing the Mie analysis. While the propagation of quantifiable uncertainties leads to an error estimate of ~25%, considering the simplifiations that were necessary for the Mie analysis, we consider a 50% closure to be an adequate agreement. Despite this, we cannot exclude additional methanol insoluble brown carbon. Conversely, the fit in Figure 6A indicates that the apparent MAC of water-soluble species was a fourth of the respective methanol MAC, according to the slope of only 12 ± 3%. Only the aged data have been fit to illustrate this point. This strong disagreement shows that the BrC in our samples was hardly water soluble, even for the most aged samples. As we expect that the majority of OA in our samples formed by wood pyrolysis (Di Blasi, 2008; Corbin et al., 2015b; Shafizadeh, 1984), we can compare our results directly to those of Chen and Bond (2010), who also found that primary wood-pyrolysis BrC was water insoluble. Moreover, the water-insoluble nature of the light absorbing components of SOA is in line with the results by Bruns et al. (2016) who showed that SOA precursors during these experiments were predominantly aromatic compounds.*

We modified the Figure 6 caption:

*Figure 6: Comparison of the $MAC_{OA}(370nm)$ of aged aerosols determined from online and offline absorption measurements. The offline filter-extraction method directly quantified properties of total OA (ordinate), while the average of $MAC_{SOA}$ and $MAC_{POA}$ weighted with respective mass concentrations is shown on the abscissa. The panels show offline measurements of (A) water-soluble OA, (B) methanol-soluble OA. Fits are to aged data only due to the significantly smaller scatter of those data, although primary data on average follow similar trends. The fitted slopes and intercepts are, respectively, (A) 0.13 ± 0.02 and 0.05 ± 0.06 $m^2 g^{-1}$ and (B) 0.12 ± 0.1 and 0.38 ± 0.03 $m^2 g^{-1}$.*

We modified the Figure S13 caption:

*Figure S13: $MAC_{OA}$ at $\lambda$ = 370 nm calculated from aethalometer measurements vs. $k_{OA}$ at $\lambda$ = 370 nm from the UV/visible measurements of the methanol extracts. The shaded region shows the 90% confidence interval of a weighted orthogonal regression (slope 66 ± 9 $m^2 g^{-1}$, intercept 0.0 ± 0.3 $m^2 g^{-1}$) to illustrate the relatively small range of variability in the data for aged samples.*

34.    L427: Fig. 6 normalizes out any uncertainty/variability in the measured [OA], because both absorption values are normalized by this. Fig. 4, in contrast, does not. How can the authors rule out the possibility that there is not some variability in the measurement of OA between burns, perhaps dependent on particle shape or variability in bounce in the AMS (which can differ between POA and SOA)?

Fig. 4 actually does normalize the absorption by [OA], so we are not sure which figure the reviewer had in mind. Our goal with Fig. 6 is to relate the offline and online absorption measurements, in such a way that any unknown uncertainties would influence scatter in the plot.

The bounce-related collection efficiency of the AMS was concluded as close to 1.0 for wood-burning OA in the literature reviewed by Corbin et al. (2015b; in their Section S1.2). From recent results from our group using a similar setup, we have measured with an AMS primary organic aerosol rich emissions from smoldering biomass and found the POA collection efficiency to be close to 1.

We reanalyzed our SMPS, AMS, and eBC (MWAA-calibrated AE33) data for the present study by fitting the SMPS mass (predicted with a density of 1.5 g/cm3) against the total PM mass predicted as AMS OA + eBC. The 95% CI of the slope of this fit corresponds to a CE of 0.7-1.0 (relative 19% relative uncertainty), consistent with the literature cited above. Combining this 19% uncertainty (slope uncertainty) with the 30% uncertainty already assigned to the AMS OA (largely reflecting uncertainties in RIE) in quadrature results in a 36% uncertainty in AMS OA, which we have updated in the text.

Shape-related collection efficiency issues in the AMS are unlikely as such issues mainly come into play when transmission through the AMS lens is considered. That is, shape itself is not an issue in the AMS except as it affects aerodynamic diameters (DeCarlo et al., 2014). In our study, particles were large enough that transmission and therefore shape-related issues were minor.

We modified the text:

> L94. *Uncertainties related to particle collection efficiency in the AMS are considered negligible for the relatively-large particles sampled here, which in terms of volume are within the size range transmitted efficiently by the AMS aerodynamic lens (Liu et al., 2007). The collection efficiency of wood-combustion OA is expected to be unity (Corbin et al., 2015b).*

We have replied to comment #26 about the data in Fig. 7 and we think we have addressed all the points raised by the reviewer in this question. We acknowledge that the Saleh and Lu data sets are not entirely independent, but have treated the corresponding parameterizations as independent for lack of any method to disentangle their interdependence. We expect other studies to cite Saleh and Lu's separately. Note that we have not focused on the more comprehensive Lu data set because the Saleh dataset represents biomass burning specifically, which we have also studied in the present work.

*Figure 7: Imaginary part of the OA refractive index at 370 nm, obtained from offline UV/vis spectroscopy of methanol OA extracts, plotted as a function of $fOA$. The data could be empirically represented by a linear function in the log-log space, in the measurement range. The ordinary least-squares fit is ($k_{OA,nm}$) = $log(M_{BC}/M_{OA})(0.51 \pm 0.07) + (-0.98 \pm 0.05)$. Also shown are parameterizations of $k_{OA}(370nm)$ for open burning against $M_{BC}/M_{OA}$ estimated based on the $k_{OA}(550nm)$ measurements in Saleh et al. (2014) and Lu et al. (2015), using the $k_{OA}$ wavelength dependence reported by the respective authors.*

We have also updated the text at line 460:

*The parameterizations reported by these authors are included in Fig. 7, where the wavelength dependence reported by those authors has been used to adjust their parameterizations to 370nm.*

40.    Fig. 7: The logic of a linear fit to the observations is not clear to me. The authors have argued that the SOA is absorbing, and differently absorbing than the POA. If I use the equation given and extrapolate towards M_BC/M_OA - >0, the k_OA -> 0. If the SOA is absorbing, and if SOA formation drives the decrease in the M_BC/M_OA, then the limiting value of k_OA should be equal to the value for k_SOA. As such, the provided fit does not seem appropriate and requires justification. Some of this may be experiment-to-experiment variability. But the limiting case issue remains.

We do agree with the reviewer that $k_{OA}$ will likely tend towards $k_{POA}$ and $k_{SOA}$ when $M_{BC}/M_{OA}$ is very large and very small, respectively. However, within the range covered, a line is the simplest model which can describe our data adequately and the linear fit used is simply empirical. We do not aim to include a complete physical interpretation in this simple fit With our fit in Fig. 7, we mainly wish to emphasize that $k_{OA}$ can be described as a function of $M_{BC}/M_{OA}$ only. For clarification we have modified the figure caption as follows:

*Figure 7: Imaginary part of the OA refractive index at 370 nm, obtained from offline UV/vis spectroscopy of methanol OA extracts, plotted as a function of **$f$OA**. The ordinary least-squares fit is **$log(k_{OA,370nm})$ = $log(M_{BC}/M_{OA})(0.51\pm0.07)$ + $(-0.98\pm0.05)$** and illustrates that the observed $k_{OA}$ can be described as a function of $M_{BC}/M_{OA}$ with reasonable accuracy, regardless of the degree of aging. Also shown are parameterizations of **$k_{OA}(370$nm)** for open burning against $M_{BC}/M_{OA}$ estimated based on the online **$k_{OA}(550$nm)** measurements in Saleh et al. (2014) and Lu et al. (2015), using the **$k_{OA}$** wavelength dependence reported by the respective authors.*

41.    Fig. 7: The authors should be able to, from their observations and within their assumptions, calculate M_POA/M_SOA. They might consider plotting k_OA vs. this ratio instead of versus M_BC. These will be related, of course, since the authors assume POA is proportional to BC during aging for a given experiment.

We agree, however, we do not think that this calculation will shed any new insights into our data set. In Fig. 7, we have only chosen to use $M_{BC}/M_{OA}$ as an abscissa because previous studies have used this quantity; our goal in Fig. 7 is a comparison of our results with related literature. We do not find this ratio to be particularly meaningful or interesting physically, but we acknowledge that future studies are likely to measure it as well and so it provides a useful basis for comparison.

42.     The origin of these "uncertainties" is unclear. They are explained later for f_OA, but for the MAC values it is not abundantly clear.

We apologize but we do not see which part of the manuscript the reviewer is referring to. We assume that the reviewer is referring to the fitted MAC values, which we have commented on above and adjusted the manuscript to include.

43. L486: This statement regarding mass yields of SOA requires much further detail.

The analysis of SOA gas phase precursors has been thoroughly presented in Bruns et al. (2016) and the discussion about SOA yields is beyond the scope of this study. As this statement is not required for the understanding of the paper we have removed it in the corrected version of the manuscript. The section now reads as follows:

> The $M_{SOAP,WLC}/M_{POA,WLC}$ was on average equal to 7.8 (GSD = 1.4) and $kOH$ was estimated as $2.7 \times 10^{-11}$ molecule$^{-1}$ cm$^3$ (GSD = 1.4), consistent with the SOA precursors chemical nature measured (e.g. PAH and phenol derivatives) by a proton-transfer-reaction mass spectrometer (PTR-MS) (Bruns et al., 2016, 2017). These high rates and enhancement ratios indicate the rapid production of SOA.

44. L512: A note about terminology. I am not certain that "error analysis" is appropriate here. Variance in the POA fraction is not "error." It is variability. A substantial aspect of this "error analysis" is really just a "sensitivity analysis." I suggest that the authors limit the term "error analysis" to when they are truly considering errors, and use some other term when they are considering variability. This is true here and elsewhere.

We agree with the reviewer and have changed the word "error" to "sensitivity".

We have also made the requested modifications related with the section entitled "*Uncertainties and variability in MAC$_{BC}$, MAC$_{POA}$ and MAC$_{SOA}$*".

45.     L499: The authors should clarify the origin of the solar irradiance data that they have used.

We have now added the reference on which the solar irradiance data are based:

[revised manuscript text omitted]

**Wall loss correction**

Solving the equations in Section 3.1 requires the determination of the time-dependent concentrations of the different absorbing species, which may be governed by their photochemical production or decay as well as by diffusion, electrostatic and gravitational losses to the walls. Assuming all particles are equally lost to the walls, an inert, non-volatile species, $X$, follows a first order decay:

$$X(t) = X(t_0) \cdot \exp(\tau^{-1}(t - t_0)) \tag{S1}$$

Here, $t$ and $t_0$ denote the time of interest and reference time, respectively. The time constant $\tau$ is the lifetime of $X$ with respect to particle wall losses. We determined $\tau$ by fitting $b_{abs}(t, 880nm)$ to Equation S1. Only the last period of each experiment was chosen for fitting, when secondary organic aerosol production rates are smaller. On average, $\tau$ equals 3.9±0.8 hours for our chamber. The wall loss corrected absorption coefficient, $b_{abs}^{WLC}(t, 880\,nm)$, varied less than 8% throughout the experiment, with higher values in the first period of measurements. Therefore we conclude that the first order decay is an appropriate approach for the wall loss correction of inert particulate properties. We ascribe the residual variations of $b_{abs}^{WLC}(t, 880\,nm)$ to a combination of uncertainties, including the aethalometer compensation parameter and possible small changes of $MAC_{BC}(880nm)$ with aging.

For the extrapolation of our data to ambient environments we computed the average SOA mass formed as a function of OH exposure during the different experiments. This step requires the correction of OA mass for particle wall losses, which has been achieved by assuming two cases: (1) condensable oxidized gases do not interact with wall-deposited particles and (2) condensable oxidized gases condense at similar rates onto the suspended and wall-deposited particles (Pierce et al., 2008). We did not consider the deposition of oxidized vapors onto the clean Teflon walls, which would require knowledge of the saturation vapor pressures of the compounds, the condensed phase bulk properties and the vapor-wall equilibration rates. It is likely that the large particle condensational sinks utilized here (with a particle surface area concentration of several hundreds of $\mu m^2$ $cm^{-3}$), outcompeted vapor deposition onto the walls. Therefore, we consider the vapor deposition to the clean Teflon wall to be of a minor importance compared to burn-to-burn variability and other experimental uncertainties.

Solving the mass balance equations of the suspended organic aerosol, $[OA_{sus}(t)]$, and the organic aerosol on the walls yields the expressions in Equations (S2) and (S3), when considering scenario (1) and (2), respectively:

$$\left[M_{OA,wlc,1}(t)\right] = \left[M_{OA,sus}(t)\right] + \int_0^t \tau^{-1}\left[M_{OA,sus}(t)\right]dt \tag{S2}$$

$$\left[M_{OA,wlc,2}(t)\right] = \left[M_{OA,sus}(t)\right] * \exp(\tau^{-1}t) \tag{S3}$$

Here, $[M_{OA,wlc}(t)]$ is the wall-loss-corrected OA concentration. The results presented in Fig. 8 in the manuscript are the average time-series of all experiments considering both scenarios, and associated ranges entail both the experiment-to-experiment variability and the uncertainties related to wall loss corrections.

**Supplementary figures**

[Figure]

**Figure S1: Absorption coefficients of fresh and aged emissions measured at 6 different wavelengths (i.e. 370 – 880 nm) using the aethalometer. The OA is measured using an AMS. Dotted lines are primary-subtracted OA (SOA) and absorption coefficient ($b_{absSOA}$(370nm)) . The black boxes mark the times where the primary, slightly aged (Aged1, OH exposure ~1x10$^7$molecules cm$^{-3}$ h ) and heavily aged filters (Aged2, OH exposure ~ 4x10$^7$ molecules cm$^{-3}$ h) were collected.**

Old Figure:

[Figure]

New Figure:

[Figure]

Figure S2: (a) Probability density function (PDF) comparing the MAC values determined by normalizing MWAA absorption measurements of offline primary (filter A), slightly aged (filter B: Aged1) and aged filter (filter C: Aged2) samples to EC (EUSAAR2) measurements of the same samples (bold line). A literature value for pure BC is also shown (Bond et al., 2006) (dashed blue line). (b) PDF comparing aethalometer attenutation measurements at 880 nm and MWAA absorption measurements at 850 nm to retrieve the aethalometer $C$ value.

[Figure]

Figure S3: Wall-loss-corrected aethalometer absorption coefficient at 880 nm normalized to start-of-experiment absorption. The lack of any trend in this plot illustrates that the wall loss correction is appropriate and that only a negligible absorption increase occurs due to additional lensing by SOA.

Old Figure:

[Figure]

New Figure:

[Figure]

**Figure S4: SMPS measurements (top), lognormal fits (middle; $f_{POA}$=0.51), and fit residuals (bottom) of the size distribution of biomass burning organic aerosol during a typical aging experiment.**

Old Figure:

[Figure]

New Figure:

[Figure]

**Figure S5: Absorption coefficients of fresh wood burning emissions measured using an aethalometer normalized to the eBC mass as a function of wavelength. In the legend each color denotes the $\alpha_{BC+POA}$(370nm,880nm) for an individual experiment. The dashed lines mark the absorption profiles calculated assuming a constant $\alpha_{BC+POA}$ in the range 370-880nm. The range of α values, is set between 1-2.2 (with an increment of 0.2), based on literature reports for primary biomass burning emissions from residential heating. The observed absorption spectra have steeper gradients with decreasing wavelength compared to the lines of constant alpha. This systematic decrease in $\alpha(\lambda, 880nm)$ with increasing $\lambda$ reflects the more-efficient light absorption by BrC at shorter wavelengths (Moosmüller et al., 2011), and shows that the power law wavelength dependence is an inaccurate oversimplification for this mixed aerosol.**

Old Figure:

[Figure]

New Figure:

[Figure]

**Figure S6: Relationship of $\alpha_{BC+POA+SOA}(\lambda, 880nm)$ to $f_{OA}$ for seven wavelengths, with symbol sizes indicating OH exposure.**

Old Figure:

[Figure]

New Figure:

[Figure]

**Figure S7: Analysis of the fitting errors of** $\alpha(\lambda, 880nm)$ **of primary emissions as a function of** $f_{OA}$. **Panel A shows the** $\alpha$ **residual as a probability density function. Panel B is an image plot of the** $\alpha(\lambda, 880nm)$ **error,** $\sigma_{\alpha(t_0,\lambda,880nm)}$, **as a function of** $f_{OA}$ **at different wavelengths.** $\sigma_{\alpha(t_0,\lambda,880nm)}$ **is obtained from the error propagation of Eq. (13) solved for different wavelengths, using the geometric mean and standard deviation of** $MAC_{POA}(\lambda)$ **and** $MAC_{BC}(\lambda)$. **This error term represents the variability in or the confidence level on the** $\alpha(t_0,\lambda, 880nm)$ **at different wavelengths. As** $\alpha(t_0,\lambda, 880nm)$ **depends on** $M_{OA}/b_{abs}(t_0, 880nm)$ **in Equation 13,** $\sigma_{\alpha(t_0,\lambda,880nm)}$ **also does. We expressed** $M_{OA}/b_{abs}(t_0, 880nm)$ **as** $f_{OA}$, **using** $\sigma_{ATN}$ **to estimate EC mass from** $b_{ATN}(880nm)$. **At short wavelengths and low OA fractions, the confidence level on** $\alpha$ **is within 0.1. However, with increasing** $f_{OA}$, **and at longer wavelength the uncertainty in predicting** $\alpha$ **increases.**

[Figure]

**Figure S8: Power law fits through the average MAC of BC, POA and SOA calculated from Aethalometer measurements plotted as a function of wavelength. Note that** $MAC_{SOA}(880nm)$ **and** $MAC_{POA}(880nm)$ **are zero by definition.**

Old Figure:

[Figure]

New Figure:

[Figure]

**Figure S9: Probability distributions of α and ln(A) describing the optical properties of BC, POA and SOA. Parameters for representing these distributions as a bivariate normal joint density function are shown in Table S1.**

The equation needed to generate these probabilities is $f(X) = \frac{|\Sigma|^{-\frac{1}{2}}}{2\pi} \exp\left(-\frac{1}{2}(X-\mu)^T \Sigma^{-1}(X-\mu)\right)$, where $\mu = \begin{pmatrix} \mu_\alpha \\ \mu_{\ln(A)} \end{pmatrix}$; represents the average α and ln(A) values and $\Sigma = \begin{pmatrix} \sigma_\alpha^2 & \rho\sigma_\alpha\sigma_{\ln(A)} \\ \rho\sigma_\alpha\sigma_{\ln(A)} & \sigma_{\ln(A)}^2 \end{pmatrix}$ is the covariance matrix. As α and ln(A) are determined from fitting the MAC vs. λ, their covariance is high. Therefore the selection of these parameters to represent the MAC profiles of BC, POA and SOA, should not be done independently but by using the probability density function above and the parameters in Table S1.

**Table S1: Parameters for normal joint density function.**

|   | BC | POA | SOA |
|---|---|---|---|
| **μ** | $\begin{pmatrix} 1.2 \\ 10.0 \end{pmatrix}$ | $\begin{pmatrix} 4.6 \\ 29.2 \end{pmatrix}$ | $\begin{pmatrix} 5.6 \\ 33.3 \end{pmatrix}$ |
| **Σ** | $\begin{pmatrix} 0.09 & 0.054 \\ 0.054 & 0.61 \end{pmatrix}$ | $\begin{pmatrix} 0.64 & 2.6 \\ 2.6 & 4.1 \end{pmatrix}$ | $\begin{pmatrix} 1.36 & 11.4 \\ 11.4 & 8.35 \end{pmatrix}$ |

Old Figure:

[Figure]

New Figure:

[Figure]

**Figure S10:** $MAC_{SOA}$ **as a function of OH exposure color coded according to the wavelength.**

[Figure]

**Figure S11: Absorbance measurements from UV-visible analysis of water extracted filters from several wood burning experiments showing very good repeatability (consistent within 10%).**

[Figure]

**Figure S12: MAC$_{bulk}$ (bulk absorbance of extracts normalized to AMS-measured OA) of primary, slightly aged (Aged1, OH exposure ~ $1 \times 10^7$ molecules cm$^{-3}$ h) and aged emissions (Aged2, OH exposure ~ $4 \times 10^7$ molecules cm$^{-3}$ h ) for (A) water and (B) methanol extracts. The bold lines indicate the medians, and the dashed lines mark the 25th and 75th percentiles. At 450–500 nm, the methanol-extract absorption shows a constant absorptivity feature which was not present before aging, suggesting that the absorbing species may be partially-oxidized (partially-solubilized) primary OA, or reflect light-absorbing SOA.**

Old Figure:

[Figure]

New Figure:

[Figure]

**Figure S13: MAC$_{OA}$ at λ = 370 nm calculated from aethalometer measurements vs. k$_{OA}$ at λ = 370 nm from the UV/visible measurements of the methanol extracts. The shaded region shows the 90% confidence interval of a weighted orthogonal regression (slope 66 ± 9 m$^2$g$^{-1}$, intercept 0.0 ± 0.3 m$^2$g$^{-1}$) to illustrate the relatively small range of variability in the data for aged samples.**

[Figure]

**Figure S14. Similar to Fig. 7 in the main text, but plotted against $f_{OA}$ for comparison to the other figures in this work.**